# Self-Directed Node Classification on Graphs

**Georgy Sokolov**[*]                                                            SOKOLOV.GM@PHYSTECH.EDU
*Moscow Institute of Physics and Technology (MIPT), Moscow, Russia*

**Maximilian Thiessen**[*]                                            MAXIMILIAN.THIESSEN@TUWIEN.AC.AT
*TU Wien, Vienna, Austria*

**Margarita Akhmejanova**[*]                                                MECHMATHRITA@GMAIL.COM
*King Abdullah University of Science and Technology (KAUST)*
*Thuwal, 23955-6900, Kingdom of Saudi Arabia*

**Fabio Vitale**                                                                FABIO.VITALE@CENTAI.EU
*CENTAI Institute, Turin, Italy*

**Francesco Orabona**                                                        FRANCESCO@ORABONA.COM
*King Abdullah University of Science and Technology (KAUST)*
*Thuwal, 23955-6900, Kingdom of Saudi Arabia*

**Editors:** Gautam Kamath and Po-Ling Loh

## Abstract

We study the problem of classifying the nodes of a given graph in the self-directed learning setup. This learning setting is a variant of online learning, where rather than an adversary determining the sequence in which nodes are presented, the learner autonomously and adaptively selects them. While self-directed learning of Euclidean halfspaces, linear functions, and general multiclass hypothesis classes was recently considered, no results previously existed specifically for self-directed node classification on graphs. In this paper, we address this problem developing efficient algorithms for it. More specifically, we focus on the case of (geodesically) convex clusters, i.e., for every two nodes sharing the same label, all nodes on every shortest path between them also share the same label. In particular, we devise an algorithm with runtime polynomial in $n$ that makes only $3(h(G) + 1)^4 \ln n$ mistakes on graphs with two convex clusters, where $n$ is the total number of nodes and $h(G)$ is the Hadwiger number, i.e., the size of the largest clique minor of the graph $G$. We also show that our algorithm is robust to the case that clusters are slightly non-convex, still achieving a mistake bound logarithmic in $n$. Finally, we devise a simple and efficient algorithm for homophilic clusters, where strongly connected nodes tend to belong to the same class.

**Keywords:** self-directed learning, online learning, node classification, graphs, time complexity.

## 1. Introduction

We study the problem of learning node clusters of a given graph $G = (V, E)$, where $V$ and $E$ respectively denote its vertex and edge set, in the *self-directed* setting (Goldman and Sloan, 1994; Ben-David et al., 1997), that recently regained novel interest (Devulapalli and Hanneke, 2024; Diakonikolas et al., 2023; Kontonis et al., 2023). Each node in $v \in V$, with $|V| = n$, is associated with a label belonging to $Y = \{1, 2, \ldots, k\} = [k]$, where $k$ is the total number of classes, through a labeling function $y : V \to Y$. In this learning setting, in an online fashion, at each time step $t \in \{1, \ldots, n\}$ the learner is required to select a node $v_t \in V$ and predict its label. After each

---

[*]Equal contribution.

prediction, the learner receives the true label $y(v_t)$, and, if it is different from the predicted one $\hat{y}(v_t)$, it made a mistake. The learner continues until all $n$ labels are predicted and received, i.e., for exactly $n$ trials. Its goal is to predict all labels while minimizing the number of mistakes.

Our work is motivated by node classification in graphs that emerge in various real-world domains, such as, social networks (including collaboration and citation networks), communication networks, biological networks (e.g., protein-protein interaction networks, RNA structures), and infrastructure networks (e.g., transportation). On this class of graphs, self-directed learning can be used to model common practical problems. For example, in advertising on a social network, the advertiser can offer one among $k$ products to each user (nodes), who can either buy it or not. Since we learn the reaction of the user(s) after each offer, it is advantageous to offer the product to a single user per trial in a sequential fashion exploiting the knowledge of the (social) network topology. This approach allows us to observe the reaction (label) and tailor subsequent offers to minimize the number of mistakes. Another example is the public transportation system, where each station is a node in a connected graph. Decisions involve allocating resources like trains to each station sequentially, one station at a time. The nodes (stations) are labeled based on the urgency of service reinforcement or type of service required. The goal is to minimize assignment errors to maintain operational efficiency and passenger satisfaction. Finally, in smart grids, each substation or component distributing energy is a node in a graph, connected to indicate the energy transmission paths. Nodes are labeled with the type of energy distributed or the demand level. Decisions on how to distribute energy to each node are made sequentially, aiming to minimize errors in energy distribution.

Even assuming the graph is fully given, learning cannot happen without an inductive bias. Hence, in Section 3, we assume that all clusters are *convex*, that is, for any pair of nodes $a, b$ in the same cluster, all nodes on shortest paths between $a$ and $b$ also belong to exactly that cluster. This assumption holds for many communities in real-world graphs, as we discuss in Section 5. In Section 3.1, we also show how to deal with labelings where this assumption is violated for some pairs of nodes. While the convexity assumption is related to the more common *homophily* assumption—the tendency of strongly connected nodes to be associated with the same class, they capture different aspects and are independent of each other. There can be convex and strongly non-homophilic clusters, and vice versa. We consider specifically homophilic clusters in Section 4.

**Main contributions.**

1. We propose the problem of self-directed node classification.

2. We devise an algorithm with polynomial runtime in $n$, called GOOD4, that learns any labeling given by two convex clusters with at most $3(h(G)+1)^4 \ln n$ mistakes, where $h(G)$ is the size of the largest clique minor of $G$ (Theorem 8).

3. We devise a robust variant of our algorithm relaxing the convexity assumption, achieving a mistake bound of $3(h(G)+1)^4 \ln n + 4M^*$, where $M^*$ is the minimum number of label flips requried to obtain a convex labeling (Section 3.1).

4. We establish general lower bounds on the number of mistakes (Section 3.2) and explore graph families for which our bounds are nearly optimal (Section 3.3).

5. For (not necessarily convex) homophilic labelings we develop a simple linear-time algorithm achieving the mistake bound $|\partial \mathcal{C}_y| + 1$ where $\partial \mathcal{C}_y$ is the cut-border induced by the true

labeling $y$, that is, all nodes that are adjacent to a node with different label (Proposition 16). We provide a related lower bound given in terms of the merging degree (Proposition 17).

**Related work.** Self-directed learning (Goldman and Sloan, 1994; Ben-David et al., 1997; Ben-David and Eiron, 1998) is a variant of the standard online learning problem (Littlestone, 1988), which allows the learner to select the points itself instead of a worst-case adversary. More recently, the mistake complexity of multiclass self-directed learning was characterized by Devulapalli and Hanneke (2024). Hanneke et al. (2023) provided mistake bounds in the related *sequence-transductive* (also called worst-case sequence) offline model, which lies in between the self-directed and the online variant. Diakonikolas et al. (2023) studied a self-directed variant of linear classification and Kontonis et al. (2023) tackled the corresponding regression problem. For node classification, active learning (Afshani et al., 2007; Guillory and Bilmes, 2009; Cesa-Bianchi et al., 2010) and online learning (Herbster et al., 2005; Cesa-Bianchi et al., 2009a, 2013; Herbster et al., 2015) were considered so far. Only Herbster et al. (2005) stated first results on a budgeted variant of self-directed learning[1] for node classification, where a fixed number of self-directed rounds are performed and afterwards learning proceeds in the usual online setup. We close this gap and provide first efficient algorithms and mistake bound for self-directed node classification.

We focus on the problem of learning clusters that are *geodesically convex*, a well-studied variant (Harary and Nieminen, 1981; Duchet and Meyniel, 1983; van de Vel, 1993; Pelayo, 2013) of standard Euclidean convexity. Seiffarth et al. (2023) studied supervised learning of such convex clusters in a graph, Bressan et al. (2021) and Thiessen and Gärtner (2021) considered the active learning, and Thiessen and Gärtner (2022) the online learning variant. Bressan et al. (2024) studied a related setting of clusters that are closed under *induced* paths.

## 2. Preliminaries

Let $G = (V, E)$ be a simple and connected graph. Here, simple refers to the fact that there is at most one edge between any pair of nodes. For the rest of the paper, we always assume the graphs to be connected and simple. Let $y : V \to Y$ be the node labels with $Y = [k] = \{1, \ldots, k\}$, where $k \in \mathbb{N}$ is the number of classes. Let $n = |V|$ for the number of nodes. For all $i \in [k]$, we call $C_i = \{v \in V \mid y(v) = i\}$ a *cluster* and let $\mathcal{C}_y = \{C_1, \ldots, C_k\}$. We call an edge $\{u, v\} \in E$ a *cut-edge* if $y(u) \neq y(v)$. A node incident to a cut-edge is a *cut-node* and the set of all cut-nodes is called the *cut-border*, denoted by $\partial \mathcal{C}_y$.

We operate within the *self-directed learning* setting, which lies between classical active and online learning. In this setting, the learner has access to the graph $G$ and does not know the labels $y$. Then, for each trial $t = 1, \ldots, n$, we execute the steps

1. Learner selects $v_t \in V \setminus \{v_1, \ldots, v_{t-1}\}$.

2. Learner predicts $\hat{y}_t(v_t) \in [k]$.

3. Learner observes $y_t(v_t) \in [k]$ and incurs a mistake if and only if $\hat{y}_t(v_t) \neq y_t(v_t)$.

The learner's goal is to minimize the number of mistakes. Let $\mathrm{M}(A, y)$ be the number of mistakes made by algorithm $A$ on node set $V$ with labeling $y$. Given a hypothesis space $\mathcal{H} \subseteq [k]^V$, we denote

---

[1] Herbster et al. (2005) call their budgeted variant of self-directed learning active learning.

by $\mathrm{M}(A, \mathcal{H}) = \max_{y \in \mathcal{H}} \mathrm{M}(A, y)$ the maximum number of mistakes $A$ over all labelings belonging to $\mathcal{H}$. For $k = 2$, we denote the VC dimension (see, e.g., Vapnik and Chervonenkis (1971); Shalev-Shwartz and Ben-David (2014)) of $\mathcal{H}$ as $\mathrm{vc}(\mathcal{H})$. If $\mathcal{H}$ is known to the learner and the true labeling $y$ belongs to the hypothesis space $\mathcal{H}$, this is known as the *realizable* setting. The (realizable) *self-directed learning complexity* of a given $\mathcal{H}$ is $\mathrm{M}(\mathcal{H}) = \min_A \mathrm{M}(A, \mathcal{H})$, i.e., the number of mistakes an optimal algorithm would make. For $k = 2$, this quantity is nicely characterized by the rank of certain game trees (Ben-David et al., 1997; Ben-David and Eiron, 1998) similarly to the *Littlestone dimension* (Littlestone, 1988) and was recently generalized to the multiclass case by Devulapalli and Hanneke (2024). We emphasize the difference to the more standard online learning on graphs protocol, where by contrast the node is adversarially selected in each step (Littlestone, 1988; Herbster et al., 2005; Cesa-Bianchi et al., 2009a).

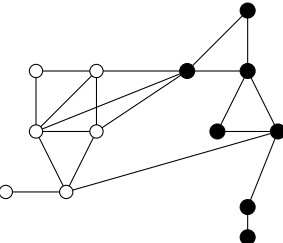

Figure 1: Example of a graph with a convex 2-labeling. In such labelings for any two nodes of the same label, all nodes on every shortest path between them also share the same label.

**Graph convexity.**   We focus on clusters that are *(geodesically) convex*, a notion closely related to ordinary convex sets in Euclidean space. A cluster $C \subseteq V$ is convex if and only if for all two nodes $a, b \in C$, also all nodes on every shortest path between $a$ and $b$ are in $C$. Note the resemblance to the standard definition of convex sets in Euclidean space through line segments.

We introduce here just the required concepts and refer the reader to Duchet and Meyniel (1983), van de Vel (1993), and Pelayo (2013). Let $I(u, v) = \{x \in V \mid x \text{ is on a shortest } u\text{-}v \text{ path}\}$ be the *(geodesic) interval* for $u, v \in V$. A set $C \subseteq V$ is convex if and only if for all $a, b \in C$, it holds that $I(a, b) \subseteq C$. Convex sets in Euclidean space can be similarly defined through $I_{\|\cdot\|_2}(u, v) = \{x \in \mathbb{R}^d \mid \|u - x\|_2 + \|x - v\|_2 = \|u - v\|_2\}$. Having defined convex sets, we can define convex hulls $\mathrm{conv}(A) = \bigcap_{C \supseteq A, C \text{ convex}} C$. If we only have two clusters and both are convex, we call the labeling a *convex bipartition* or *halfspace* of the graph, see Figure 1 for an example. One main subject of study in convexity theory are *separation axioms*, $S_1$ to $S_4$, which characterize the separation ability of halfspaces (Bandelt, 1989; Chepoi, 1994, 2024). We will only use the $S_4$ separation axiom. We say a graph is $S_4$ (i.e., it satisfies $S_4$) if for any pair $A, B \subseteq V$ with $\mathrm{conv}(A) \cap \mathrm{conv}(B) = \emptyset$ there exists a halfspace $H \subseteq V$ such that $A \subseteq H$ and $B \subseteq V \setminus H$.

**Clique number, Hadwiger number, and treewidth.**   We denote by $K_w$ the *clique* (or *complete*) graph on $w \in \mathbb{N}$ nodes. The *clique number* $\omega(G)$ of a graph $G$, is the size of the largest clique that is a subgraph of $G$. A graph $H$ is a *minor* of another graph $G$ if $H$ can be derived from $G$ by a sequence of edge contractions, edge deletions, and vertex deletions, see Tutte (1961), Robertson and Seymour (1985), and an example in Figure 2. Here, contracting an edge means merging the two nodes connected by the edge and removing the edge itself. We denote by $h(G)$ the *Hadwiger number* of $G$, which is the size of the largest clique minor in $G$. Thus, if $G$ is minor-free for some

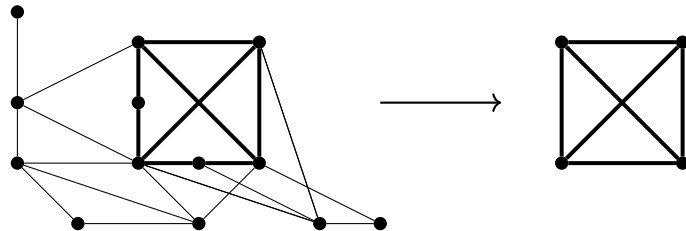

Figure 2: Example of a graph with a $K_4$-minor

$K_w$, it holds that $h(G) < w$. Intuitively $h(G)$ is a measure of sparsity. Another such measure is the treewidth $\mathrm{tw}(G)$, quantifying the tree-likeness of $G$. For a definition of treewidth see Robertson and Seymour (1984, 1986).

## 3. Self-directed learning of convex bipartitions

In this section, we introduce our main contribution, a polynomial time algorithm for self-directed learning of halfspaces $\mathcal{H}$ of $G$ achieving a near-optimal mistake bound. Full proofs can be found in the appendix. Before we discuss our proposed algorithm, let us briefly consider existing algorithms and their drawbacks.

**Proposition 1 (Ben-David et al. 1997)** *Let $\mathcal{H} \subseteq 2^V$ be a hypothesis space with $|V| = n$. Then, it holds that*

$$\Omega\left(\frac{\mathrm{vc}(\mathcal{H})}{\ln n}\right) \leq \mathrm{M}(\mathcal{H}) \leq \mathcal{O}(\mathrm{vc}(\mathcal{H}) \ln n).$$

The upper bound $\mathcal{O}(\mathrm{vc}(\mathcal{H}) \ln n)$ can easily be achieved by the HALVING algorithm (Littlestone, 1988). Thus, in the case of halfspaces on a graph, HALVING achieves the following mistake bound.

**Proposition 2** *Let $G = (V, E)$ be a graph with $n = |V|$ and let $\mathcal{H}$ be the set of convex bipartitions of $G$. Then, $\mathrm{M}(\mathrm{HALVING}, \mathcal{H}) = \mathcal{O}(h(G) \ln n)$.*

However, naively running HALVING requires a majority vote over the version space, i.e., the set of all hypotheses that are consistent with the data so far, in each step. Unfortunately, computing the version space for geodesic halfspaces is known to be NP-hard (Seiffarth et al., 2023). The optimal algorithms of Devulapalli and Hanneke (2024) and Ben-David et al. (1997) for self-directed learning have similar issues as they depend on the version space, as well. More generally, it is not known if such bounds can be achieved by a computational efficient algorithm, i.e., with polynomial computational complexity.

Here, we propose a new algorithm, GOOD4 (Good Quadruples), that has a better trade-off between computational runtime complexity and the number of mistakes: GOOD4 runs in polynomial time and achieves a mistake bound similar to HALVING.

Before presenting the actual algorithm let us start with some intuition and observations. We use the fact that for large enough subsets of nodes in sparse graphs, shortest paths intersect. For that, let us define a *quadruple* as a set $\{(a, b), (c, d)\}$ containing two pairs of nodes $(a, b)$ and $(c, d)$, where $a$, $b$, $c$, $d$, are four distinct nodes. We define a quadruple as a *good quadruple* if there exists one shortest path connecting $a$ to $b$ that intersects at least one shortest path connecting $c$ to $d$, that is,

they share at least one common node. Said differently, $I(a,b) \cap I(c,d) \neq \emptyset$. A sufficient condition for the existence of many good quadruples is that the Hadwiger number $h(G)$ is not too large.

**Proposition 3** *Let $G$ be $K_w$-minor free (i.e., $h(G) < w$). Then, any subset of $\max(w,4)$ nodes contains a good quadruple.*

We care about good quadruples $\{(a,b),(c,d)\}$, since they cannot be labeled arbitrarily, e.g., $a, b \in C_1$ and $c, d \in C_2$ is not possible as otherwise one of the two clusters would not be convex. For a set $U \subseteq V$, we denote by $Q(U)$ the set of quadruples in the $U$ and by $Q_{\text{good}}(U)$ the set of good quadruples in $U$. By $q(U)$ and $q_{\text{good}}(U)$ we denote the size of the respective set. Next, we show that good quadruples exist and that there is a significant number of them for large enough $U$.

**Observation 4** *Let $G = (V,E)$ be a $K_w$ minor-free graph. For any subset $U \subseteq V$ of size at least $\max(w,4)$ nodes, the relative number of good quadruples $q_{\text{good}}(U)/q(U)$ in $U$ is at least $8/w^4$.*

Next, we define good nodes. The intuition is that good nodes participate in many good quadruples with many other nodes. These are the nodes which we are going to select for prediction.

**Definition 5** *($U$-good nodes and $\varepsilon_U$) Let $U \subseteq V$. Define*

$$\varepsilon_U = \frac{q_{\text{good}}(U)}{8q(U)} \ .$$

*A node $a \in U$ is a $U$-good node, if there exists a subset $U_a \subseteq U$ of size at least $\lceil 4\varepsilon_U(|U|-1)\rceil$, such that for all $b \in U_a$, the number of pairs $c, d \in U$ resulting in good quadruples $\{(a,b),(c,d)\}$ is at least*

$$\left\lceil 4\varepsilon_U \binom{|U|-2}{2} \right\rceil \ .$$

*If $U$ is clear from context, we might simply say $a$ is a good node and also write $\varepsilon = \varepsilon_U$.*

In other words, for a good node $a$ and for each $b \in U_a$ it holds that the shortest paths between $a$ and $b$ intersect with a $\frac{q_{\text{good}}(U)}{8q(U)}$ fraction of shortest paths with endpoints in $U$. Next, we now show that there exists at least one good node for each large enough set $U$.

**Observation 6** *Let $G$ be $K_w$ minor-free. Then, in any subset $U \subseteq V$ of at least $\max(w,4)$ nodes, each node participating in the maximum number of good quadruples is $U$-good.*

This leads to the following observation.

**Observation 7** *Let $G$ be a $K_w$ minor-free graph and $U \subseteq V$ a set with at least $\max(w,4)$ nodes. Then, for a good node $a \in U$ and every $b$ in $U_a$, there exist at least $\lceil \varepsilon(|U|-2)\rceil$ nodes $c' \in U$ such that for at least $\lceil \varepsilon(|U|-3)\rceil$ of the nodes $d' \in U$, $\{(a,b),(c,d)\}$ forms a good quadruple.*

Our high-level idea is as follows. Using $U$-good nodes we want to either learn a large number of node labels without mistakes or on mistake discard a $\varepsilon$-fraction of good quadruples. That way we can employ a Halving-like strategy on the set of all quadruples.

The main loop of GOOD4 (Algorithm 1) continues as long as there are good quadruples in the set of unlabeled nodes $U$, i.e., $|Q_{\text{good}}(U)| > 0$. Each iteration of this loop involves the following four main steps, constructing good quadruples $\{(a,b),(c,d)\}$. **Step 1: find a good node $a$.** We

---

**Algorithm 1:** GOOD4 Algorithm

---

**input:** Graph $G = (V, E)$

1   $U \leftarrow V$

2   **while** $|Q_{\text{good}}(U)| > 0$ **do**

     /\* Step 1:  Find a good node $a$ \*/

3      **for** $a' \in U$ **do** $Q^{a'}_{\text{good}}(U) \leftarrow \{\{b, c, d\} : \{(a', b), (c, d)\} \in Q_{\text{good}}(U)\}$

4      $a \leftarrow \arg\max_{a' \in U} \left| Q^{a'}_{\text{good}} \right|$

5      **predict** arbitrarily $\hat{y}(a)$;    **observe** $y(a)$;   $\widetilde{y} \leftarrow y(a)$;   $U \leftarrow U \setminus \{a\}$

     /\* Step 2:  Find corresponding node $b$ \*/

6      mistake $\leftarrow$ False

7      **for** $b' \in U$ **do** $Q^{a,b'}_{\text{good}}(U) \leftarrow \{(c, d) : \{(a, b'), (c, d)\} \in Q_{\text{good}}(U \cup \{a\})\}$

8      **while** mistake = False **and** $|U| > 0$ **do**

9          $b \leftarrow \arg\max_{b' \in U} \left| Q^{a,b'}_{\text{good}} \right|$

10         **predict** $\hat{y}(b) = 1 - \widetilde{y}$;    **observe** $y(b)$;   $U \leftarrow U \setminus \{b\}$

11        **if** $\hat{y}(b) \neq y(b)$ **then** mistake $\leftarrow$ True

12      **if** mistake = False **then** **continue**

     /\* Step 3:  Find corresponding node $c$ \*/

13      mistake $\leftarrow$ False

14      **for** $c' \in U$ **do** $Q^{a,b,c'}_{\text{good}}(U) \leftarrow \{d : \{(a, b), (c, d)\} \in Q_{\text{good}}(U \cup \{a, b\})\}$

15      **while** mistake = False **and** $|U| > 0$ **do**

16         $c \leftarrow \arg\max_{c' \in U} \left| Q^{a,b,c'}_{\text{good}} \right|$

17         **predict** $\hat{y}(c) = \widetilde{y}$;    **observe** $y(c)$;   $U \leftarrow U \setminus \{c\}$

18         **if** $\hat{y}(c) \neq y(c)$ **then** mistake $\leftarrow$ True

19      **if** mistake = False **then continue**

     /\* Step 4:  Iterate over all corresponding $d$ \*/

20      **for** $d \in U$ *s.t.* $\{(a, b), (c, d)\} \in Q_{\text{good}}(U \cup \{a, b, c\})$ **do**

21         **predict** $\hat{y}(d) = \widetilde{y}$;    **observe** $y(d)$;   $U \leftarrow U \setminus \{d\}$

22         **if** $\hat{y}(d) \neq y(d)$ **then break**

   /\* Step 5:  Predict remaining labels \*/

23  **predict** arbitrary labels for any remaining nodes in $U$

---

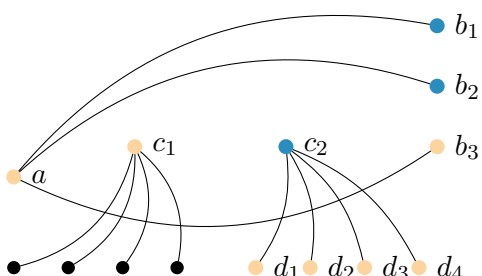

Figure 3: An example illustrating how GOOD4 operates. Here curves denote shortest paths and two crossing curves are a good quadruple. Using the good quadruples $\{(a, b_3), (c_2, d_i)\}$ for $i \in \{1, \ldots, 4\}$ we can infer the labels of the nodes $d_i$.

find a good node $a$, predict an arbitrary label for it and observe the true label $y(a)$. **Step 2: find a corresponding node** $b$**.** The algorithm processes each node in $U$ and finds the node $b$ that together with $a$ participates in the maximum number of good quadruples $\{(a, b), (c, d)\}$. The algorithm predicts for $b$ the opposite label of $a$, $\hat{y}(b) = 1 - y(a)$, observes the true label $y(b)$, and removes $b$ from $U$ (see Figure 3 for an example). The algorithm repeats this step as long as we do not make any mistake and $U$ is not empty. This process is referred to as *selecting nodes $b'$ in decreasing order*. **Step 3: find a corresponding node** $c$**.** If the prediction for any node $b$ was incorrect, the algorithm searches within $U$ for another node $c$ which together with $a$ and this node $b$ participates in the maximum number of good quadruples $\{(a, b), (c, d)\}$. The algorithm predicts $\hat{y}(c) = y(a)$ for $c$, observes the true label $y(c)$, and removes $c$ from $U$. The algorithm repeats this step while $\hat{y}(c) = y(c)$ and $U$ is not empty. This process is referred to as *selecting nodes $c'$ in decreasing order*. If $U$ becomes empty we exit from the algorithm. **Step 4: predict labels for all such quadruples.** If the prediction for any node $c$ was incorrect, the algorithm predicts the label $y(a)$ for each node $d$, where $\{(a, b), (c, d)\}$ forms a good quadruple, learns the true label of $d$ and removes $d$ from $U$. If we made a prediction mistake we stop with this step. **Step 5: predict remaining labels.** If no more good quadruples exist while the graph is still not fully labeled, we predict arbitrary labels for the remaining nodes in $U$. This corresponds to the case when the induced subgraph $G[U]$ is exactly a complete graph $K_{|U|}$ or $|U| \leq 3$.

Overall, this leads to our main result, which is the following bound on the number of mistakes of GOOD4 for learning halfspaces.

**Theorem 8 (Mistake upper bound)** *Let $G = (V, E)$ be a graph with $n = |V|$ and let $\mathcal{H}$ be the set of convex bipartitions of $G$. Then, GOOD4 (Algorithm 1) runs in polynomial time in $n$ and makes* $\mathrm{M}(\mathrm{GOOD4}, \mathcal{H}) \leq 3(h(G) + 1)^4 \ln n$ *mistakes.*

We also developed a multiclass variant of GOOD4 leading again to an algorithm with $\mathrm{poly}(n)$ runtime and at most a logarithmic number of mistakes in $n$ for fixed $h(G)$, see Appendix D.

**Correctness.** Here, we give an overview of the proof, for a full analysis see Appendix B.2. By Observation 4 we know that $q_{\mathrm{good}}(U)/q(U)$ is at least $\frac{8}{(h(G)+1)^4}$ for a sufficiently large $U$. Next, using Observation 6, we see that $U$-good node exist for sufficiently large $U$ and

$$\varepsilon = \frac{q_{\mathrm{good}}(U)}{8q(U)} \geq \frac{1}{(h(G) + 1)^4} .$$

As $a$ is a good node, there is a large number of nodes $b'$, such that there are at least $4\varepsilon\binom{|U|-2}{2}$ good quadruples $\{(a, b'), (c, d)\}$. In **Step 1**, we then simply find a node $a$ participating in the maximum number of good quadruples, which has to be $U$-good (for large enough $U$). In **Step 2**, we either discover at least $\varepsilon|U|$ labels or make a mistake on a node $b$ and move to Step 3. In the latter case as we selected the nodes $b'$ in decreasing order, the node $b$ and the $\lceil 4\varepsilon(|U| - 1)\rceil$ selected first all belong to the set $U_a$ from Theorem 5. Therefore, by Observation 7 for $b$, there exist at least $\lceil\varepsilon(|U|-2)\rceil$ nodes $c$, such that for at least $\lceil\varepsilon(|U|-3)\rceil$ of the nodes $d$, the quadruple $\{(a, b), (c, d)\}$ is good. In **Step 3**, we take nodes $c'$ in decreasing order. If we do not make mistakes among the first $\lceil\varepsilon(|U| - 2)\rceil$ nodes $c'$, we discover at least $\varepsilon|U|$ labels in total; otherwise, we find a node $c$ among the first where we made a prediction mistake and go to Step 4. In **Step 4**, we use this node $c$. We know that there are many nodes $d'$ such that $\{(a, b), (c, d')\}$ forms a good quadruple. So, we predict label $y(a)$ for these $d'$. By convexity, we will make no mistake on all such nodes. Finally, we conclude that in total, at least $\varepsilon|U|$ node labels were inferred during these four steps. Thus, we overall have $\mathcal{O}(\log_{1-\varepsilon}(n)) = \mathcal{O}(h(G)^4 \ln n)$ iterations of the while loop.

### 3.1. Learning near-convex labelings

In real-world scenarios, it is unrealistic to expect the node labeling to be convex. Even if in particular tasks we can expect convex labelings, deviations from the ideal convex bipartition are always possible. Therefore, let us consider a setting where the node labeling is not convex, but, in some sense, close to convex. To measure the deviation from convexity, we introduce a very natural concept. We call the labeling $M^*$-*near-convex* if it can be converted into a convex labeling by flipping at most $M^*$ nodes. This is related to agnostic online learning (Ben-David et al., 2009).

**Theorem 9** GOOD4 *can predict all labels in $G$ with at most $4M^* + 3(h(G) + 1)^4 \ln n$ mistakes, where $M^*$ is the smallest integer such that the labeling of $G$ is $M^*$-near-convex.*

We emphasize that we do not need to know $M^*$ in advance.

### 3.2. Lower bounds

Let us first show that the dependence on the Hadwiger number, without relying on further graph parameters, is unavoidable in general.

**Proposition 10** *For any $h, n \in \mathbb{N}$ with $h \le n$, there exists a graph $G$ with $h(G) = h$, $|V| = n$, and convex bipartitions $\mathcal{H}$, such that $\mathrm{M}(\mathcal{H}) = \Omega(h)$.*

For $S_4$ graphs we get the following lower bound.

**Proposition 11** *Let $G$ be an $S_4$ graph with $n$ nodes. Then, any algorithm learning convex bipartitions of $G$ will make $\Omega\left(\frac{\omega(G)}{\ln n}\right)$ in the worst-case.*

Thus, for the broad family of $S_4$ graphs, we see that the mistake bound is largely determined by the denseness of the graph, here in terms of the clique number $\omega(G)$. Only the gap between the clique number and Hadwiger number remains. As we discuss in Section 3.3, this gap is typically quite small for many graph families, such as chordal or bounded treewidth graphs.

### 3.3. Bounds for specific graph families

We discuss some broad families of graphs, where we achieve near-optimal mistake bounds.

*Bounded treewidth graphs.* Treewidth of a graph is a measure of "tree-likeness" and gives an upper bound on the Hadwiger number $h(G) \leq \text{tw}(G) + 1$. Many common graph families have bounded treewidth. For example, trees, $k$-outerplanar graphs, Halin graphs, and series-parallel graphs all have constant treewidth (see, e.g., Bodlaender, 1993) and hence also constant Hadwiger number. Thus, on all such graphs we will make only $\mathcal{O}(\ln n)$ many mistakes using GOOD4. Hyperbolic random graphs have a treewidth of $\mathcal{O}\left((\ln n)^2\right)$ with high probability if the degree of the power law is chosen as $\beta \geq 3$ (Bläsius et al., 2016). Here, we achieve a mistake bound of $\mathcal{O}\left((\ln n)^3\right)$.

*Planar graphs.* Planar graphs are a well known and broad graph family. They have Hadwiger number at most 4. For planar graphs we can actually enumerate all halfspaces in polynomial time using the algorithm of Glantz and Meyerhenke (2017). This allows to run HALVING in polynomial time and achieve a near-optimal mistake bound $\mathcal{O}(\ln n)$. As the Hadwiger number is a constant for planar graphs our algorithm GOOD4 achieves the same bound. The downside of the approach of Glantz and Meyerhenke (2017) is that it is particularly tailored towards planar graphs and it seems rather non-trivial to generalize it to more general families. Our algorithm GOOD4 can be applied on all graphs, and in particular, on many near-planar graph families such as apex graphs it still achieves a mistake bound of $\mathcal{O}(\ln n)$.

*Graphs with $h(G) \approx \omega(G)$.* A graph $G$ is chordal if $G$ contains no induced cycles of size four or larger. Chordal graphs form a large graph family where $\omega(G) = h(G)$.

**Proposition 12** *Let $G$ be chordal. Then, $\omega(G) = h(G)$.*

Another example are *circular-arc graphs*, the intersection graphs of arcs on a circle, where $h(G) \leq 2\omega(G)$ (Narayanaswamy et al., 2007).

**Corollary 13** *Let $G = (V, E)$ be a chordal graph or circular-arc graph with $n = |V|$ and let $\mathcal{H}$ be the convex bipartitions of $G$. Then, GOOD4 makes $M(\text{GOOD4}, \mathcal{H}) = \mathcal{O}(\omega(G)^4 \ln n)$ mistakes.*

We see that the mistake complexity is largely determined by $\omega(G)$ for chordal graphs and by combining this with Proposition 11, we see that this dependence is also necessary in general.

*Bipartite graphs.* Bipartite graphs are graphs without cycles of odd length.

**Proposition 14** *Let $G$ be a bipartite graph and $\mathcal{H}$ its convex bipartitions. We can learn any labeling in $\mathcal{H}$ with at most 2 mistakes in linear time.*

For the special case of grid graphs (Cartesian product of two paths), we have the following tight result even for $k$ convex classes.

**Proposition 15** *Let $G$ be any grid graph and $\mathcal{H}$ be the set of all its convex $k$-partitions, with $k \geq 2$. Thus, we have $M(\mathcal{H}) \geq k/4$ and $M(\text{GRIDWALKER}, \mathcal{H}) \leq 3k$ while GRIDWALKER runs in time linear in $n$.*

## 4. Learning homophilic labelings

A common alternative assumption to the convexity of the clusters $\mathcal{C}_y$ is to assume that they are homophilic, see e.g., Herbster et al. (2005); Cesa-Bianchi et al. (2013); Dasarathy et al. (2015). In

this section we focus on such graphs, quantifying homophily by assuming that the size of the cut-border $|\partial \mathcal{C}_y|$ is small. In this setting, the following simple graph traversing strategy gives a bound of $|\partial \mathcal{C}_y| + 1$.

**Proposition 16** *Let $G = (V, E)$ be a graph with $n = |V|$ and $m = |E|$. Then, there exists an algorithm* TRAVERSE *that learns in total linear time $\mathcal{O}(n + m)$ any (not necessarily convex) labeling $y \in [k]^V$ with at most $|\partial \mathcal{C}_y| + 1$ mistakes.*

To achieve a related lower bound we adapt a proof by Cesa-Bianchi et al. (2009b, 2011), which holds for a different variant of the online learning setting. The lower bound is in terms of the *merging degree*, another complexity measure of the cut-border. Here, we use a different definition of *clusters*. Let a cluster be any *maximal* connected subgraph of $G$ that is *uniformly* labeled. Note that with this definition we can have up to $n$ clusters even when $k = 2$. Given any cluster $C$, we denote by $\partial C$ its cut-border, by $\underline{\partial C} = \partial C \cap C$ its *inner border*, and by $\overline{\partial C} = \partial C \setminus \underline{\partial C}$ its *outer border* of $C$. Finally, the merging degree $\delta(C)$ of $C$ is then defined as $\delta(C) = \min(|\underline{\partial C}|, |\overline{\partial C}|)$. The merging degree of the whole graph $G$, is defined as $\delta(G) = \sum_{C \in \mathcal{P}_y} \delta(C)$, where $\mathcal{P}_y$ is the partition into the clusters induced by $y$.

**Proposition 17** *Given any graph $G$ and any integer $c < n$, there exists a labeling $y$ satisfying $|\delta(G)| \leq 2c$ such that any algorithm makes at least $c$ mistakes.*

Thus, for $k = 2$ and two connected clusters $C_1$, $C_2$, the algorithm TRAVERSE achieves a near-optimal mistake bound, as long as the cut-border is balanced, that is, the inner borders of $C_1$ and $C_2$ have roughly the same size.

## 5. Discussion

In this section, we compare the self-directed learning setting with other learning settings, discuss whether real-world graphs fit our assumptions, and state interesting directions.

**Comparison with other learning models.** Let us compare our bounds to previously known results in active and online learning. In *active learning* the goal is to learn with a small number of queries instead of the number of mistakes; essentially we skip step 2 of our setup. It follows that the self-directed learning mistake complexity is always smaller than the number of queries. In general, there can be arbitrarily large gaps between the number of self-directed mistakes the number of queries. For example, already on tree graphs the number of queries is linear in the number of leaves, while the number of self-directed mistakes is at most 2.

Related issues arise in *online learning*, where an adversary chooses the nodes (in step 1 of our setup) to be labeled instead of the learner. On one hand by Theorem 1 we know that the online and self-directed mistake complexities are at most a $\mathcal{O}\left((\ln n)^2\right)$ factor apart. On the other hand, the best known efficient algorithm for halfspaces (Thiessen and Gärtner, 2022) has an online mistake bound that depends on the largest $\ell$ such that the complete bipartite graph $K_{2,\ell}$ is a minor of $G$, a quantity typically much larger than the Hadwiger number. This term can be linear in $n$ for planar and bounded treewidth graphs, while we achieve a logarithmic number of self-directed mistakes.

**Hadwiger number and sparsity of real-world graphs.** Many notions to quantify the sparsity of graphs, besides Hadwiger number, exist (Nešetřil and De Mendez, 2012; Demaine et al., 2019). One of the most studied assumptions is that the treewidth $\text{tw}(G)$ of the graph is constant. For such graphs, our polynomial time algorithm GOOD4 achieves a logarithmic number of mistakes in $n = |V|$ as $h(G) \leq \text{tw}(G) + 1$. Indeed, many real-world graphs have small to moderate treewidth. For example, communication networks (de Montgolfier et al., 2011), infrastructure based networks (Maniu et al., 2019), and social contracts on blockchains (Chatterjee et al., 2019) tend to have a small treewidth. Hyperbolic random graphs as discussed before have a small treewidth for certain parameter choices and are often used to model social networks (Bläsius et al., 2016). Also in biology many networks with small treewidth arise, such as proteins (Peng et al., 2015) and protein-protein-interaction networks (Blanchette et al., 2012). RNA structures are known to typically have a treewidth below 6 (Song et al., 2005). Molecules tend to have treewidth 2 or 3. For example, the molecules in the datasets NCI and PubChem (of size 250k and 135k molecules) have treewidth at most 3 (Horváth and Ramon, 2010; Böcker et al., 2011). Also 92-98% of molecules from common small and large-scale molecular benchmark datasets have treewidth at most 2 (Bause et al., 2025).

**Convexity of clusters in real-world graphs.** While the most common bias for communities in graphs is homophily, there is recent interest to develop graph learning approaches explicitly for non-homophilic data (Lim et al., 2021). One potentially appropriate, alternative bias is our considered assumption of convex or near-convex clusters. For example, Thiessen and Gärtner (2021) showed that the majority of communities in large-scale real-world networks (like DBLP, YouTube, and Amazon products) are indeed convex. Furthermore, Marc and Šubelj (2018) and Šubelj et al. (2019) showed that infrastructure and collaboration networks have a "tree of cliques"-like structure leading to connected subgraphs being convex. Also in biology convex clusters arise, e.g., in gene-similarity (Zhou et al., 2002) and protein interaction networks (Li et al., 2012).

**Open problems.** This is the first paper on self-directed learning on graphs, so naturally several questions remains unsolved. First of all, it is unclear if our lower bound for learning convex bipartitions is tight. In particular, we do not know if it is possible to design algorithms whose mistake bound depends polynomially on $\omega(G)$ instead of $h(G)$ like in GOOD4. We think that the optimal mistake bound for convex bipartitions might be $\omega(G) + 1$. For example, the claim holds for weakly median graphs (Thiessen and Gärtner, 2021). However, even if an algorithm exists that make at most these numbers of mistakes, we do not believe that such algorithms can be implemented in polynomial time (e.g., due to the hardness of enumerating the version space (Seiffarth et al., 2023)). Similar questions hold for the multiclass case, in addition to establishing if a better analysis of multiclass GOOD4, or a better algorithm, is possible. Finally, it would be valuable to identify a broad and significant class of input graphs for which our algorithm achieves optimality with respect to the merging degree discussed in Section 4.

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

# Appendix A. Graphs with no big clique minor

Our main algorithm GOOD4 is designed to operate on graphs which do not contain complete graphs $K_w$ as a minor, where $w$ is a parameter. We describe how the algorithm efficiently restores all labels in convex 2-labeling within these graphs with a maximum of $3w^4 \ln n$ mistakes, all while operating within polynomial time in $n$. Then, we show a mistake bound of $4M^* + 3w^4 \ln n$, if the labeling can be made convex after flipping the labels of at most $M^*$ nodes.

Before introducing the algorithm, let us review our notation and discuss some properties of graphs without complete graph $K_w$ as a minor. A graph $H$ is a minor of another graph $G$ if $H$ can be derived from $G$ by a sequence of edge contractions, edge deletions, and removal of isolated nodes, see an example in Figure 2. In this context, contracting an edge means merging the two nodes connected by the edge and removing the edge itself. Now, suppose $G = (V, E)$ does not contain any complete graph $K_w$ as a minor. Denote by $U \subseteq V$ the *unknown set*, consisting of all nodes whose labels we do not know. In a graph $G$, we define a *quadruple* as a set $\{(a, b), (c, d)\}$ containing two unordered pairs of nodes, $(a, b)$ and $(c, d)$, where $a$, $b$, $c$, and $d$ are four distinct nodes from the unknown set $U$. We emphasize that for the sake of notational simplicity we use $(a, b)$ for unordered pairs in the context of quadruples, i.e., $(a, b) = (b, a)$. We define a quadruple as a *good quadruple* if the shortest paths connecting $a$ to $b$ and $c$ to $d$ intersect, that is, they share at least one common node. If there are multiple shortest paths between $a$ and $b$ or between $c$ and $d$, then at least one shortest path from each pair should intersect. Let $Q(U)$ denote the set of all quadruples in $U$ and $q(U)$ its size. Similarly, let $Q_{\text{good}}(U)$ and $q_{\text{good}}(U)$ denote the set of all good quadruples and their size, respectively.

We now list a number of propositions and observations that will simplify our proof.

**Proposition 3** *Let $G$ be $K_w$-minor free (i.e., $h(G) < w$). Then, any subset of $\max(w, 4)$ nodes contains a good quadruple.*

**Proof** If this were not the case we could consequently contract all edges on the shortest paths between each pair of given $w$ nodes and receive a clique $K_w$ which contradicts the assumption that $G$ is $K_w$-minor-free. ∎

**Observation 4** *Let $G = (V, E)$ be a $K_w$ minor-free graph. For any subset $U \subseteq V$ of size at least $\max(w, 4)$ nodes, the relative number of good quadruples $q_{\text{good}}(U)/q(U)$ in $U$ is at least $8/w^4$.*

**Proof** The total number of all quadruples is exactly

$$\frac{1}{2}\binom{|U|}{2}\binom{|U| - 2}{2} = 3\binom{|U|}{4} = \frac{|U|(|U| - 1)(|U| - 2)(|U| - 3)}{8}.$$

For $w > 4$, consider the set of all possible combinations of $w$ distinct nodes from $U$. By Proposition 3, each set of $w > 4$ nodes contains a good quadruple. On the other hand, each good quadruple can be extended in $\binom{|U|-4}{w-4}$ ways up to the set of $w$ distinct nodes. Hence, the total number of good quadruples, is at least $\frac{\binom{|U|}{w}}{\binom{|U|-4}{w-4}} = \frac{|U|(|U|-1)(|U|-2)(|U|-3)}{w(w-1)(w-2)(w-3)} > \frac{|U|^4}{w^4}$ and the ratio of good quadruples to the total number of all quadruples is at least $\frac{8}{w^4}$. ∎

**Observation 6** *Let $G$ be $K_w$ minor-free. Then, in any subset $U \subseteq V$ of at least $\max(w, 4)$ nodes, each node participating in the maximum number of good quadruples is $U$-good.*

**Proof** Given that each quadruple contains four nodes and there are $q_{\text{good}}(U)$ good quadruples, it follows that some node $a'$ participates in at least $\left\lceil \frac{4q_{\text{good}}(U)}{|U|} \right\rceil$ good quadruples. Let $a$ denote the node that participates in the most number of good quadruples. We claim that $a$ is a $U$-good node. We prove it by contradiction. If $a$ is not $U$-good, then the size of $U_a$ can be written as $\lfloor 4\varepsilon(|U| - 1) - t \rfloor$, where $t \geq 0$. Also, for the remaining $|U| - \lfloor (4\varepsilon(|U| - 1) - t) \rfloor - 1$ nodes the number of pairs $c, d \in U$ such that the quadruple $\{(a, b), (c, d)\}$ is considered good is strictly less than $\left\lceil 4\varepsilon \binom{|U|-2}{2} \right\rceil$. Hence, the number of good quadruples containing $a$ cannot exceed $\lfloor (4\varepsilon(|U| - 1) - t) \rfloor \binom{|U|-2}{2} + (|U| - \lfloor (4\varepsilon(|U| - 1) - t) \rfloor - 1) \left\lfloor 4\varepsilon \binom{|U|-2}{2} \right\rfloor$. By using Observation 4, we see that this number is strictly less than $\left\lceil \frac{4q_{\text{good}}(U)}{|U|} \right\rceil$. However, this leads to a contradiction since $a$ is required to participate in at least the same amount of good quadruples as $a'$. Therefore, a good node $a$ exists. ∎

**Observation 7** *Let $G$ be a $K_w$ minor-free graph and $U \subseteq V$ a set with at least $\max(w, 4)$ nodes. Then, for a good node $a \in U$ and every $b$ in $U_a$, there exist at least $\lceil \varepsilon(|U| - 2) \rceil$ nodes $c' \in U$ such that for at least $\lceil \varepsilon(|U| - 3) \rceil$ of the nodes $d' \in U$, $\{(a, b), (c, d)\}$ forms a good quadruple.*

**Proof** We prove it by contradiction. If there are not many $c$ for which there are many $d$, the number of good quadruples containing given good $a$ and fixed $b$ from $b \in U_a$ cannot exceed

$$\lfloor (\varepsilon(|U| - 2) - t) \rfloor (|U| - 3) + (|U| - \lfloor (\varepsilon(|U| - 2) - t) \rfloor - 2) \lfloor \varepsilon(|U| - 3) \rfloor$$

where $t \geq 0$. Note that this expression is strictly less than $4\varepsilon \binom{|U|-2}{2}$. This leads to a contradiction, as the number of good quadruples containing the given good node $a$ and the fixed $b \in U_a$ is at least $\left\lceil 4\varepsilon \binom{|U|-2}{2} \right\rceil$ by the definition of $U$-good nodes (Definition 5). ∎

## Appendix B. Binary classification

Here we describe our main algorithm, GOOD4.

### B.1. GOOD4 algorithm description

GOOD4 iteratively reduces the number of nodes with unknown labels by predicting the labels of nodes in good quadruples.

Initially, the algorithm takes as input a simple connected graph $G = (V, E)$. It initializes the set of nodes with unknown labels $U$ as $V$. The function $Q_{\text{good}}(U)$ is defined to output the set of all good quadruples in $U$. The main loop of the algorithm continues as long as $|U| > 0$. At each iteration of this main loop, if there are no good quadruples in the current set $U$, i.e., $|Q_{\text{good}}(U)| = 0$, the algorithm predicts labels for all remaining nodes in $U$ arbitrarily and terminates. Note that the case $|Q_{\text{good}}(U)| = 0$ corresponds to the situation when induced subgraph $G[U]$ of the graph $G$ is exactly a complete graph $K_{|U|}$. Each iteration of this loop involves the following four steps.

**Step 1: find a good node** $a$**.** For each node $a' \in U$ the algorithm computes the number of good quadruples $\{(a', b), (c, d)\}$ in $U$ containing $a'$ and then finds the node $a$ that participates in the maximum number of these good quadruples. The algorithm arbitrarily predicts the label $\hat{y}(a)$ for $a$, ($\hat{y}(a)$ can be either 0 or 1), observes the true label $y(a)$, stores $y(a)$ in $\widetilde{y}$, and removes $a$ from $U$.

**Step 2: find a corresponding node** $b$**.** The algorithm processes each node $b'$ in the remaining set $U$, computes the number of good quadruples $\{(a, b'), (c, d)\}$ in $U$ containing couple $(a, b')$ and then finds the node $b$ which together with $a$ participates in the maximum number of these good quadruples. The algorithm predicts the opposite label $\hat{y}(b) = 1 - \widetilde{b}$ for $b$, observes the true label $y(b)$, and removes $b$ from $U$, see Figure 3 for an example. The algorithm repeats this step while $\hat{y}(b') = y(b')$ and $U$ is not empty. This process is referred to as *selecting nodes* $b'$ *in decreasing order*. If $U$ becomes empty we exit from the algorithm, because we predicted all labels. Otherwise, we encountered a mistake and found a node $b$ with the same label as node $a$. In this case, we go to Step 3.

**Step 3: find a corresponding node** $c$**.** If the prediction for a node $b$ is incorrect, the algorithm processes each node $c'$ in the remaining set $U$, computes the number of good quadruples $\{(a, b), (c', d)\}$ in $U$ containing both couple $(a, b)$ and $c'$, and then finds the node $c$ which together with couple $(a, b')$ participates in the maximum number of these good quadruples. The algorithm repeats this step while $\hat{y}(c') = y(c')$ and $U$ is not empty. This process is referred to as *selecting nodes* $c'$ *in decreasing order*. If $U$ becomes empty we exit from the algorithm.

**Step 4: find a corresponding node** $d$**.** If the prediction for a node $c$ is incorrect, the algorithm predicts the label $\widetilde{y}$ to each node $d'$, where $\{(a, b), (c, d')\}$ forms a good quadruple, learns the true label of $d'$ and removes $d'$ from $U$. This step continues until either an incorrect prediction for $d'$ occurs or the set of such $d'$ becomes empty.

**Step 5: predict remaining labels.** If no more good quadruples exist while the graph is still not fully labeled, we predict arbitrary labels for the remaining nodes in $U$. This corresponds to the case when the induced subgraph $G[U]$ is exactly a complete graph $K_{|U|}$ or $|U| \leq 3$.

## B.2. Analysis of the algorithm

The algorithm GOOD4 operates on a connected graph $G = (V, E)$ with $n$ vertices. We will show in Theorem 8 that GOOD4 makes $\mathcal{O}(\ln n)$ mistakes for graphs with constant Hadwiger number $h(G)$. Note that the algorithm does not require knowledge of the Hadwiger number $h(G)$ beforehand. We established Observation 4, which states that in any sufficiently large $U \subseteq V$, the number of good quadruples $q_{\text{good}}(U)$ in $U$ is at least $\frac{8}{(h(G)+1)^4}$ of the total number of all quadruples $q(U)$ in $U$. Next, for $\varepsilon = \frac{q_{\text{good}}(U)}{8q(U)}$ (which is at least $\frac{1}{(h(G)+1)^4}$), we introduced the notion of a $U$-good node and proved Observation 6, which confirms the existence of such nodes in any sufficiently large $U \subseteq V$. We also showed that the node $a$ participating in the most number of good quadruples is $U$-good. Hence, selecting $a$ in **Step 1** of the algorithm ensures $a$ is a $U$-good node provided that $|U|$ is large enough. Assuming $|U| > \lceil \frac{1}{\varepsilon} \rceil + 3$ guarantees that $|U| > h(G) + 1$ and that Observations 4,6, and 7 hold. Note that if $|U|$ becomes smaller, the number of mistakes made by the algorithm is trivially at most $|U|$.

In **Step 2** we either discover at least $\varepsilon|U|$ labels or, by making one mistake in Step 2, move to Step 3. Consider the second case. Since we selected the nodes $b'$ in decreasing order and $a$ is a $U$-good node, the node $b$ where we made a mistake as well as other first $\lceil 4\varepsilon(|U|-1) \rceil$ nodes selecting in accordance of this decreasing order, belongs to the set $U_a$, where $U_a$ is from Definition 5. Therefore,

we may use Observation 7 to state that for this $y$, there exist at least $\lceil \varepsilon(|U| - 2)\rceil$ nodes $c'$ such that for at least $\lceil \varepsilon(|U| - 3)\rceil$ of the nodes $d'$, such as $\{(a, b), (c', d')\}$ forms a good quadruple. Note that we do not claim that the labels of all $c'$ and $d'$ nodes are unknown, because some of them might have already been observed during Step 2 when we checked nodes $b'$ in decreasing order.

In **Step 3**, we take $c'$ in decreasing order (line 16), ensuring we have many $d'$. If we do not make mistakes among the first $\lceil \varepsilon(|U| - 2)\rceil$ nodes $c'$, we discover at least $\varepsilon|U|$ labels in total; otherwise, we find a node $c$ among the first $\lceil \varepsilon(|U| - 2)\rceil$ nodes which label we predicted with a mistake.

In **Step 4**, we operate with this wrongly predicted node $c$. We know that there are many $d'$ such that $\{(a, b), (c, d')\}$ forms a good quadruple. So, we predict label $\widetilde{y}$ for these $d'$. It is impossible for a vertex $d'$ in Step 4 to be processed with a mistake. In such a scenario, both $a$ and $b$ would have the label $\widetilde{y}$, while $c$ and $v$ would have the opposite label, $1 - \widetilde{y}$, which is impossible when the labeling is convex. Finally, we conclude that in total, at least $\varepsilon|U|$ labels were predicted during these four steps.

To conclude, in **Step 5**, given that there are no more good quadruples, we have that $|U| \leq h(G)$ and in the worst case we commit a mistake on all the remaining nodes. Now we are ready to state and prove Theorem 8.

**Theorem 8 (Mistake upper bound)** *Let $G = (V, E)$ be a graph with $n = |V|$ and let $\mathcal{H}$ be the set of convex bipartitions of $G$. Then, GOOD4 (Algorithm 1) runs in polynomial time in $n$ and makes $\mathrm{M}(\mathrm{GOOD4}, \mathcal{H}) \leq 3(h(G) + 1)^4 \ln n$ mistakes.*

**Proof** We prove the mistake bound and discuss the runtime separately.

*Mistake bound.* In each round $i$ of GOOD4, we fix a good node $a$ and learn at least an $\varepsilon$-fraction of the labels from the current unknown set $U$, making no more than three mistakes. This process incrementally reduces the size $\sigma$ of the unknown set $U$ at each round, that we represent as the sequence

$$\sigma_0 = |V|, \sigma_1, \ldots, \sigma_q,$$

where $q$ is the total number of rounds after the very first one, such that $\sigma_q \geq \lceil \frac{1}{\varepsilon}\rceil + 3$ holds. Note that $\sigma_i \leq \sigma_{i-1}(1 - \varepsilon)$ holds for each $i \in [q]$. Hence,

$$\sigma_q \leq (1 - \varepsilon)\sigma_{q-1} \leq \ldots \leq (1 - \varepsilon)^q \sigma_0 = (1 - \varepsilon)^q n.$$

Consequently, the number of rounds $q$ needed after the very first one is bounded by $\lceil \log_{\frac{1}{1-\varepsilon}} \frac{n}{\frac{1}{\varepsilon}+3}\rceil$. The total number of mistakes made is thus upper bounded by $3\lceil \log_{\frac{1}{1-\varepsilon}} \frac{n}{\frac{1}{\varepsilon}+3}\rceil + \lceil \frac{1}{\varepsilon}\rceil + 3$. Note that by Observation 4, the ratio between good quadruples $q_{\mathrm{good}}(U)$ and the number of all quadruples $q(U)$ in any sufficiently large subset $U$ of $K_{h(G)+1}$-minor-free graphs is at least $\frac{8}{(h(G)+1)^4}$. Hence, we have $\varepsilon = \frac{q_{\mathrm{good}}(U)}{8q(U)} \geq \frac{1}{(h(G)+1)^4}$. Also, note that

$$3\left\lceil \log_{\frac{1}{1-\varepsilon}} \frac{n}{\frac{1}{\varepsilon} + 3}\right\rceil + \left\lceil \frac{1}{\varepsilon}\right\rceil + 3 \leq 3\frac{\ln \frac{n}{\frac{1}{\varepsilon}+3}}{\ln \frac{1}{1-\varepsilon}} + \frac{1}{\varepsilon} + 7\,,$$

and the right-side is decreasing in $\varepsilon$ on the interval $(0, 1)$. Hence, the number of mistakes does not exceed

$$\frac{3 \ln\left(\frac{n}{(h(G)+1)^4+3}\right)}{\ln\left(\frac{(h(G)+1)^4}{((h(G)+1)^4-1)}\right)} + (h(G) + 1)^4 + 7\,.$$

Finally, since

$$\ln \frac{(h(G)+1)^4}{(h(G)+1)^4 - 1} = \ln \left( 1 + \frac{1}{(h(G)+1)^4 - 1} \right)$$
$$> \frac{1}{(h(G)+1)^4 - 1} - \frac{1}{2((h(G)+1)^4 - 1)^2} > \frac{1}{(h(G)+1)^4},$$

the number of mistakes is less than

$$3(h(G)+1)^4 \ln \frac{n}{(h(G)+1)^4 + 3} + (h(G)+1)^4 + 7 =$$
$$3(h(G)+1)^4 (\ln n - \ln((h(G)+1)^4 + 3)) + (h(G)+1)^4 + 7 \le 3(h(G)+1)^4 \ln n.$$

*Runtime.* Note that we can compute the set of all good quadruples in $V$ in $\mathcal{O}(n^4 d)$ time. This bound can be derived as follows: first, by using a BFS search, we compute a shortest path for each pair of vertices, which takes $\mathcal{O}(n^4)$ time. Then, for each of the pairs of shortest paths processed, we verify whether they intersect in $\mathcal{O}(n)$ time. Once we found the set of all good quadruples in $V$ all other operations in the algorithm such as finding all good quadruples in $U$ for some $U$, finding all good quadruples in $U$ containing a given vertex $a$, pair $(a,b)$, pair $(a,b)$ and node $c$ as well as finding $a$, pair $(a,b)$, pair $(a,b)$ and node $c$ which maximize the number of good quadruples containing them, are linear in the size of the set of all good quadruples in $V$. ∎

## Appendix C. Learning near-convex binary labelings

In real-world scenarios, it is unrealistic to expect that node labeling will be convex. Even if the requirements of a particular task dictate a convex labeling, deviations and errors from the ideal scenario are always possible. Therefore, let us consider a scenario where the node labeling is not convex, but is, in some sense, close to being convex. To measure the deviation from convexity, we introduce a very natural concept. A labeling of the nodes is said to be $M^*$-*near-convex* if can be converted into a convex labeling by flipping no more than $M^*$ nodes. Here "flipping" refers to changing a node's label from one state to another in binary labeling. Denote by $\mathcal{M}^*$ this subset of $M^*$ nodes.

**Theorem 9** GOOD4 *can predict all labels in $G$ with at most $4M^* + 3(h(G)+1)^4 \ln n$ mistakes, where $M^*$ is the smallest integer such that the labeling of $G$ is $M^*$-near-convex.*

**Proof** Consider a quadruple $\{(a,b),(c,d)\}$ as *violating convexity* if both $a$ and $b$ are labeled 0, while $c$ and $d$ are labeled 1, or vice versa. By the definition of $\mathcal{M}^*$, at least one node among $\{a,b,c,d\}$ should belong to $\mathcal{M}^*$. The algorithm exits from the main while-loop, when

1. it either predicts with at most three mistakes at least $\varepsilon$-fraction of unknown nodes,

2. or with no more than four mistakes, it finds in $U$ a violating convexity quadruple $\{(a,b),(c,v)\}$ and then removes it from $U$. In this case, the size of $\mathcal{M}^*$ is decreasing at least by one.

In each round $i$ of GOOD4, we fix a good node $a$ and either learn at least an $\varepsilon$-fraction of the labels from the current unknown set $U$ or the size of $\mathcal{M}^*$ decreases at least by one. This process incrementally reduces the size $\sigma$ of the unknown set $U$ at each round, that we represent as the sequence

$$\sigma_0 = |V|, \sigma_1, \ldots, \sigma_q,$$

where $q$ is the total number of rounds after the very first one, such that the condition $\sigma_q \geq \left\lceil \frac{1}{\varepsilon} \right\rceil + 3$ holds. For each $i \in [q]$, we have that $\sigma_{i-1} \leq (1 - \varepsilon)\sigma_i$ or $M^*$ decreases at least by one. Consequently, the mistake bound is less than $4M^* + 3h^4 \ln n$, where $3(h(G) + 1)^4 \ln n$ is derived in the same way as in the proof of Theorem 8. ∎

## Appendix D. Multiclass classification

### D.1. Algorithm description

FINDDISTINCTLABEL algorithm is an auxiliary algorithm designed to identify nodes whose label differs from the labels of other nodes in a given set $S$. The algorithm inputs are graph $G$, set of nodes $S$, and set of labels (colors) $Z$. FINDDISTINCTLABEL starts by checking if $Z$ has only one element ($|Z| = 1$). In this scenario, it processes each node $a'$ in $S$. If the label of $a'$ is unknown, it assigns the single label $z$ to each node $a'$ in $S$, predicting $\hat{y}(a') = z$ and observing the actual label $y(a')$. Regardless of whether the label was previously known or just predicted, if $y(a')$ differs from $z$, FINDDISTINCTLABEL$(G, S, Z)$ exits and returns that node $a'$; otherwise, it continues to the next node. If all nodes match $z$, it returns $-1$.

If the set of labels $Z$ contains more than one label, the algorithm proceeds to the main part. It computes $\varepsilon = \frac{|Q_{\text{good}}(S)|}{8|Q(S)|}$. For each node $a' \in S$, it identifies good quadruples containing $a'$. Then, it selects the node $a$, which participates in the most number of good quadruples among all $a' \in S$. If the label of $a$ is unknown, it predicts an arbitrary label $\hat{y}(a)$ from $Z$ and observes the true label $y(a)$. Regardless of whether the label $y(a)$ was previously known or just predicted, if $y(a) \notin Z$, FINDDISTINCTLABEL$(G, S, Z)$ exits and returns that node $a$.

Next, the set $Y_a$ is updated to include nodes $b'$ for which there are enough good quadruples $\{(a, b'), (c, d)\}$. Namely, there are at least

$$\lceil 2\varepsilon(|S| - 2)(|S| - 3) \rceil$$

pairs $(c, d)$ such that $\{(a, b'), (c, d)\}$ is a good quadruple. The algorithm then recursively calls the FINDDISTINCTLABEL function with graph $G$, set $Y_a$, and $Z \setminus \{y(a)\}$, see Figure 4 for an example. The variable $b$ is assigned the result of the FINDDISTINCTLABEL function. This process continues recursively, going deeper and incrementing $Z$. As the recursion unwinds, the algorithm checks each result step-by-step. During the unwinding, the algorithm verifies the outcomes of the recursive calls: if it encounters $-1$, it returns $-1$; otherwise, it checks if a current node $a$ is found with a label $y(a) = y(b)$. Once such node $a$ is found, the algorithm starts working with the set $Y_{ab}$.

The set $Y_{ab}$ is updated to include nodes $c' \in S$ for which there are enough good quadruples $\{(a, b), (c', d)\}$. The algorithm recursively calls the FINDDISTINCTLABEL function with graph $G$, set $Y_{ab}$, and $\{y(a)\}$. The variable $c$ is assigned the result of the FINDDISTINCTLABEL function. If the recursive call returns $-1$, the algorithm returns $-1$. If the label of node $c$ found in the recursive

---

**Algorithm 2:** Function FindDistinctLabel

---

**input:** Graph $G = (V, E)$, set of nodes $S$, set of labels $Z$

1 **Function** `FindDistinctLabel`$(G, S, Z)$**:**

2     **if** $|Z| = 1$ **then**

3        $z \leftarrow$ the only element in $Z$

4        **for** $a' \in S$ **do**

5           **if** $y(a')$ *is unknown* **then**

6              **predict** $\hat{y}(a') = z$;    **observe** $y(a')$

7           **if** $y(a') \neq z$ // applying this step to all nodes regardless of whether the label was predicted just now or was already known

8           **then**

9              **return** $a'$

10        **return** *-1* // all nodes in $S$ predicted correctly

11     $\varepsilon \leftarrow \frac{|Q_{\mathrm{good}}(S)|}{8|Q(S)|}$

12     **for** $a' \in S$ **do**

13        $Q_{\mathrm{good}}^{a'}(S) \leftarrow \{(b, c, d) \mid \{(a', b), (c, d)\}$ is a good quadruple$\}$

14     $a \leftarrow \arg\max_{a' \in S} |Q_{\mathrm{good}}^{a}|$ // find node that is in most good quadruples

15     **if** $y(a)$ *is unknown* **then**

16        **predict** arbitrary $\hat{y}(a)$ from $Z$;    **observe** $y(a)$

17     **if** $y(a) \notin Z$ **then return** $a$

18     $Y_a \leftarrow \{b' \in S \mid$ there are $\geq$ $\lceil 2\varepsilon(|S| - 2)(|S| - 3)\rceil$ pairs $(c, d)$ such that $\{(a, b'), (c, d)\}$ is a good quadruple$\}$
       // update $Y_a$, applying this step to node $a$ regardless of whether the label was predicted just now or was already known

19     $b \leftarrow$ `FindDistinctLabel`$(G, Y_a, Z \setminus \{y(a)\})$ // recursive call

20     **if** $b = -1$ // many nodes from $Y_a$ predicted correctly

21     **then return** *-1*

22     **if** $y(b) \neq y(a)$ // thus found node with label not in $Z$

23     **then return** $b$

24     $Y_{ab} \leftarrow \{c' \in S \mid$ there are at least $\lceil \varepsilon(|S| - 3)\rceil$ nodes $d$ such that $\{(a, b), (c', d)\}$ is a good quadruple$\}$
       $c \leftarrow$ `FindDistinctLabel`$(G, Y_{ab}, \{y(a)\})$

25     **if** $c = -1$ // all nodes in $Y_{ab}$ predicted correctly

26     **then return** *-1*

27     **if** $y(c) \notin Z$ **then return** $c$

28     $Y_{abc} \leftarrow \{d' \in S \mid \{(a, b), (c, d')\}$ is a good quadruple$\}$

29     $d \leftarrow$ `FindDistinctLabel`$(G, Y_{abc}, Z \setminus \{y(c)\})$

30     **if** $d = -1$ // a big fraction of nodes from $Y_{abc}$ predicted correctly

31     **then return** *-1*

32     **if** $y(d) = y(c)$ // labeling is not convex

33     **then return** *-1*

34     **return** $d$ // else $y(d)$ is not in $Z \setminus \{y(c)\}$ and $y(d) \neq y(c)$, so $y(d)$ is not in $Z$

---

---

**Algorithm 3:** Multiclass-GOOD4

---

**input:** Graph $G = (V, E)$, set of labels $Z$

1   $U \leftarrow V$
2   **while** $|Q_{\text{good}}(U)| > 0$ **do**
3      FindDistinctLabel$(G, U, Z)$
4      remove from $U$ all nodes with known labels
5   **predict** arbitrary labels for any remaining nodes in $U$

---

call is not $y(a)$, the algorithm returns this node $c$. Once such node $c$ is found, the algorithm starts working with the set $Y_{abc}$.

The set $Y_{abc}$ is updated to include nodes $d' \in S$ for which there are enough good quadruples $\{(a, b), (c, d')\}$. Then, it recursively calls the FINDDISTINCTLABEL function with the graph $G$, the set $Y_{abc}$, and $Z \setminus \{y(c)\}$. Note that $Z$ here is decreasing. The variable $d$ is assigned the result of the FINDDISTINCTLABEL function. If the recursive call returns $-1$, the algorithm returns $-1$. If the label of the node $d$ found in the recursive call is the same as $y(c)$, the algorithm returns $-1$. If the label of node $d$ differs from $y(c)$ and is not in $Z$, the algorithm returns node $d$.

FINDDISTINCTLABEL$(G, S, Z)$ might take the same nodes multiple times—for instance, a node might be selected as $b'$, later than $c'$, and then as $d'$. But the algorithm predicts only unknown labels, thanks to a conditional check for label status.

MULTICLASS-GOOD4 predicts labels for nodes in a graph $G = (V, E)$ with a given set of labels $Z$. It starts by initializing set $U$ to contain all nodes in $V$, indicating that initially, all node labels are unknown. The algorithm then enters a while loop that continues as long as there are good quadruples in $U$. Within the loop, the function FINDDISTINCTLABEL is called with graph $G$, set $U$, and labels $Z$. This function tries to find a node in $U$ whose label is distinct from the labels in $Z$. After each call to FINDDISTINCTLABEL, the algorithm removes nodes with known labels from $U$. Once the loop terminates, indicating that there are no more good quadruples, the algorithm assigns arbitrary labels to any remaining nodes in $U$.

### D.1.1. CORRECTNESS.

**Proposition 18** FINDDISTINCTLABEL$(G, S, Z)$ *makes at most* $3 \cdot 2^{|Z|-1} - 2$ *mistakes and*

- *either finds and returns node in $S$, which label does not belong to $Z$,*

- *either reveals at least $\varepsilon^{|Z|-1}|S|$ labels in $S$ and returns $-1$,*

- *either finds in $S$ a quadruple that violates convexity, reveals all labels of these four vertices, and returns $-1$.*

**Proof** We prove the statement by induction on the size of $|Z|$.

    **Base case:** For $|Z| = 1$, there is only one label, the algorithm makes no more than one mistake and returns either node in $S$, which label does not belong to $Z$, either reveals at least $|S|$ labels in $S$ and returns $-1$. The statement is true.

    **Inductive step:** Assume the statement holds for $|Z| = k$. We must show that it also holds for $|Z| = k + 1$.

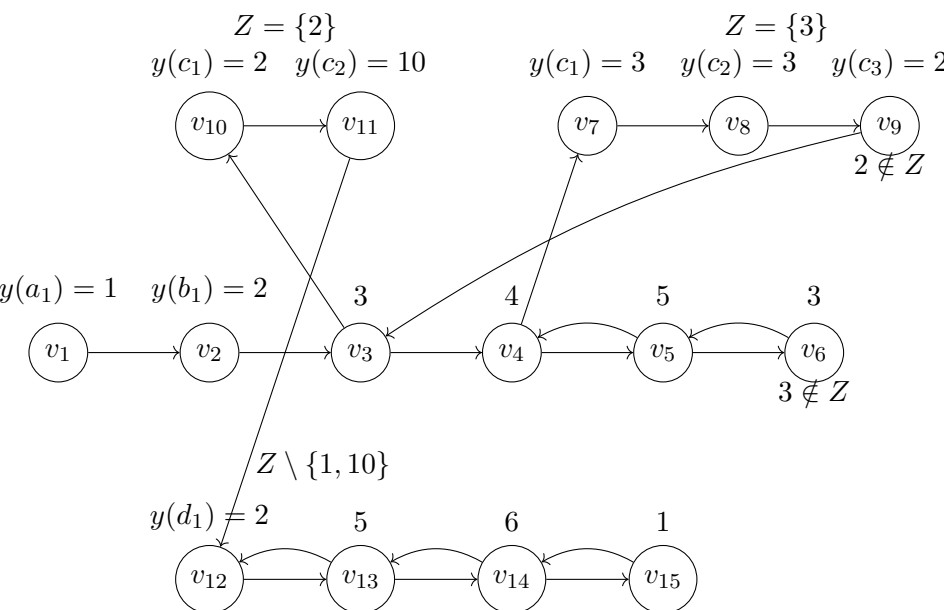

Figure 4: An example illustrating how FINDDISTINCTLABEL algorithm operates.

We go through all the points where the algorithm might terminate and verify that it has accomplished what was claimed and which of the three possibilities was fulfilled.

If we terminate at $y(a) \notin Z$, line 17, then the algorithm returned a node with a label not in $Z$ (because we just checked that in the 'if' statement) and the number of mistakes is at most 1.

If we terminate at $b = -1$, line 21, then by the induction hypothesis, the recursive call either already has found a violated convexity quadruple (in which case we can return $-1$, as a violated convexity quadruple is found), or it has determined at least $\varepsilon^{|Z|-1}|Y_a|$ labels. Using the fact that $|Y_a| \geq \varepsilon|S|$ due to Observation 6 and the definition of $U$-good node, we get that at least $\varepsilon^{|Z|}|S|$ labels were determined, and we can return $-1$. The number of mistakes is at most $1+(3 \cdot 2^{k-1}-2) = 3 \cdot 2^{k-1} - 1$.

If we terminate at line $y(b) \neq y(a)$, then the algorithm returned a node with a label not in $Z$ and the number of mistakes is at most $1 + (3 \cdot 2^{k-1} - 2) = 3 \cdot 2^{k-1} - 1$.

If we terminate at $c = -1$, line 26, then by the induction hypothesis, the recursive call either already has found a violated convexity quadruple (in which case we can return $-1$, as a violated convexity quadruple is found), or it has determined at least $\varepsilon^{|Z|-1}|Y_{ab}| + 2$ labels. Using the fact that $|Y_{ab}| \geq \varepsilon(|S| - 2)$ due to Observation 7, we get that at least $\varepsilon^{|Z|}|S|$ labels were determined, and we can return $-1$. The number of mistakes is at most $1 + (3 \cdot 2^{k-1} - 2) = 3 \cdot 2^{k-1} - 1$.

If we terminate at $y(c) \notin Z$, line 27, then the algorithm returned a node with a label not in $Z$ and the number of mistakes is at most $1 + (3 \cdot 2^{k-1} - 2) + 1 = 3 \cdot 2^{k-1}$.

If we terminate at $d = -1$, line 31, then by the induction hypothesis, the recursive call either already has found a violated convexity quadruple (in which case we can return $-1$, as a violated convexity quadruple is found), or it has determined at least $\varepsilon^{|Z|-1}|Y_{abc}| + 3$ labels. Using the fact that $|Y_{abc}| \geq \varepsilon(|S| - 3)$ due to Observation 7, we get that at least $\varepsilon^{|Z|}|S|$ labels were determined, and we can return $-1$. The number of mistakes is at most $1 + (3 \cdot 2^{k-1} - 2) + 1 = 3 \cdot 2^{k-1}$.

If we terminate at $y(d) = y(c)$, line 33, then found a violated convexity quadruple (in which case we can return $-1$, as a violated convexity quadruple is found). The number of mistakes is at most $1 + (3 \cdot 2^{k-1} - 2) + 1 + (3 \cdot 2^{k-1} - 2) = 3 \cdot 2^k - 2$.

If we return $d$, line 34, then the algorithm returned a node with a label not in $Z$ and the number of mistakes is at most $1 + (3 \cdot 2^{k-1} - 2) + 1 + (3 \cdot 2^{k-1} - 2) = 3 \cdot 2^k - 2$.

∎

We established Observation 4, which states that in any sufficiently large $S \subseteq V$, the number of good quadruples $q_{\text{good}}(S)$ in $S$ is at least $8/(h(G) + 1)^4$ of the total number of all quadruples $q(S)$ in $S$. Next, for $\varepsilon = q_{\text{good}}(S)/(8q(S))$ (which is at least $1/(h(G) + 1)^4$), we introduced the notion of a $U$-good node and proved Observation 6, which confirms the existence of such nodes in any sufficiently large $S \subseteq V$. We also showed that the node $a$ participating in the most number of good quadruples is $U$-good. Hence, selecting $a$, the algorithm ensures $a$ is a $U$-good node provided that $|U|$ is large enough. The algorithm recursively calls itself no more than $|Z|$ times to find a node $b$ such that $y(b) \notin Z$. If a such node $b$ is not found, then a big fraction of labels is predicted correctly, because we checked all nodes in the current set $Y_a$ and $Y_a$ is big enough due to Observation 6. If a such $b$ is found, then propagated back through all levels of recursion the algorithm finds a node $a$ such as $y(a) = y(b)$. Note that such node $a$ exists since $y(b)$ does not belong to the current set $Z$, but was at initial set $Z$. The algorithm then searches for a node $c \in Y_{ab}$ with $y(c) \neq y(a)$. If such a node $c$ is found, the algorithm starts a new recursive call with depth at most $Z - 1$. Since our Observation 7 guarantees that each of $Y_{a,b,c}, Y_{ab}$ and $Y_a$ include at least $\varepsilon|S|$ nodes and we proved Proposition 18, it can be concluded that for each round of MULTICLASS-GOOD4, at least $\varepsilon^k|U|$ nodes become predicted provided $U$ is big enough.

Now note that $|U| \geq \left(\left\lceil \frac{1}{\varepsilon} \right\rceil + 3\right)^k$ and Proposition 18 guarantee that if FINDDISTINCTLABEL returns $-1$ and the labeling is convex, then at least $\varepsilon^{|Z|-1}|S|$ labels have been observed after the current recursion call with set $S$. So, the size $|S|$ remains larger than $\left\lceil \frac{1}{\varepsilon} \right\rceil + 3 \geq h(G) + 1$ in all recursion levels (lines 19,24,29). Hence, Observation 4, Observation 6, and Observation 7 hold. Note that if $|U|$ becomes smaller, the number of mistakes made by the algorithm is trivially at most $|U|$.

FINDDISTINCTLABEL always ends evaluations due to Proposition 18 and that both sets $S$ and $Z$ are bounded. On the other hand, Algorithm MULTICLASS-GOOD4 always ends because the size of $U$ is bounded and after each round it removes at least $(h(G) + 1)^{-k}|U|$ nodes when $|U| \geq \left(\left\lceil \frac{1}{\varepsilon} \right\rceil + 3\right)^k$.

**Theorem 19** *Let $G = (V, E)$ be a graph with $n$ nodes and Hadwiger number $h(G)$. Suppose the $k$-labeling of nodes in $G$ is convex, where $k$ is a constant. Then, there exists a version of GOOD4, that can restore all labels in $G$ with polynomial runtime in $n$ and $\mathcal{O}(2^k(h(G))^{4k} \ln n)$ mistakes.*

**Proof** In our setting the number of clusters is $|Z| = k$. At the end of each round $i$ of MULTICLASS-GOOD4, the algorithm learns at least $\varepsilon^k|U|$ labels from the unknown set $U$ or finds four nodes that violate convexity. In the first case, the algorithm makes no more than $2^k$ mistakes, for details see Section D.1.1. The second case is impossible because the labeling is convex. This process reduces the size of the unknown set $U$, leading to a sequence of sizes of the unknown set $U$:

$$\sigma_0 = |V|, \sigma_1, \ldots, \sigma_q,$$

where $q$ is the total number of rounds after the very first one, such that $\sigma_q \geq \left(\lceil \frac{1}{\varepsilon} \rceil + 3\right)^k$ holds. We also have that $\sigma_i \leq \sigma_{i-1}(1 - \varepsilon^k)$ holds for each $i \in [q]$. Hence,

$$\sigma_q \leq (1 - \varepsilon^k)\sigma_{q-1} \leq \ldots \leq (1 - \varepsilon^k)^q \sigma_0 = (1 - \varepsilon^k)^q n.$$

Consequently, the number of rounds needed is $\mathcal{O}(\log_{1/(1-\varepsilon^k)} n) = \mathcal{O}\left(\frac{\ln n}{\varepsilon^k}\right)$. The total number of mistakes made is thus $\mathcal{O}\left(2^k \left(\frac{\ln n}{\varepsilon^k}\right)\right) + \mathcal{O}\left(\left(\lceil \frac{1}{\varepsilon} \rceil + 3\right)^k\right) = \mathcal{O}(2^k (h(G))^{4k} \ln n)$. $\blacksquare$

## Appendix E. More missing proofs

**Proposition 2** *Let $G = (V, E)$ be a graph with $n = |V|$ and let $\mathcal{H}$ be the set of convex bipartitions of $G$. Then, $\mathrm{M}(\mathrm{HALVING}, \mathcal{H}) = \mathcal{O}(h(G) \ln n)$.*

**Proof** The claim immediately follows by combing Proposition 1 and the fact that $\mathrm{vc}(\mathcal{H}) = \mathcal{O}(h(G))$ for halfspaces $\mathcal{H}$ (Duchet and Meyniel, 1983; Thiessen and Gärtner, 2021). $\blacksquare$

### E.1. Lower bounds

**Proposition 10** *For any $h, n \in \mathbb{N}$ with $h \leq n$, there exists a graph $G$ with $h(G) = h$, $|V| = n$, and convex bipartitions $\mathcal{H}$, such that $\mathrm{M}(\mathcal{H}) = \Omega(h)$.*

**Proof** Take the clique on $h$ nodes and a path with $n - h$ nodes attached to one of the nodes of the clique. If the path belongs to one cluster, we can force $h$ mistakes on the clique. $\blacksquare$

**Proposition 11** *Let $G$ be an $S_4$ graph with $n$ nodes. Then, any algorithm learning convex bipartitions of $G$ will make $\Omega\left(\frac{\omega(G)}{\ln n}\right)$ in the worst-case.*

**Proof** Let $\mathcal{H}$ be the set of halfspaces of $G$. It is well known that any algorithm will make $\Omega\left(\frac{\mathrm{vc}(\mathcal{H})}{\ln n}\right)$ in the worst-case (Ben-David et al., 1997). Additionally, it is easy to see that in $S_4$ graphs any clique is shatterable Hence, $\mathrm{vc}(\mathcal{H}) \geq \omega(G)$. $\blacksquare$

### E.2. Homophilic labelings

**Proposition 16** *Let $G = (V, E)$ be a graph with $n = |V|$ and $m = |E|$. Then, there exists an algorithm $\mathrm{TRAVERSE}$ that learns in total linear time $\mathcal{O}(n + m)$ any (not necessarily convex) labeling $y \in [k]^V$ with at most $|\partial \mathcal{C}_y| + 1$ mistakes.*

**Proof** $\mathrm{TRAVERSE}$ follows the following strategy. We run a basic graph traversing algorithm, such as, BFS or DFS. We predict an arbitrary label for the first node $v_1 \in V$. Then, for all $t \geq 2$, every time a new node is visited it is a neighbour $v_t \in V \setminus \{v_1, \ldots, v_{t-1}\}$ of a previously explored neighbour $v \in \{v_1, \ldots, v_{t-1}\}$ of $v_t$. This is the case as $G$ is connected. For $v_t$ we predict $\hat{y}_t = y(v)$ (we already know $y(v)$). We will only make a mistake if $v_t$ is a cut-node, resulting in the claimed bound. As $\mathrm{TRAVERSE}$ runs just one graph traversal overall to learn $y$ the runtime follows. $\blacksquare$

We now show an adaptation of a lower bound presented by Cesa-Bianchi et al. (2009b, 2011) which holds for a different variant of the online learning setting, when $k = 2$. The mistake bound is expressed as a function of a complexity measure called *merging degree*, which intuitively is related to the cut-border $\mathcal{C}_y$. We now recall the definition of this measure provided by Cesa-Bianchi et al. (2009b, 2011). For ease of explanation, in this lower bound we use a definition of cluster that is different from the one provided for the elements of $\mathcal{C}_y$: Let a cluster be any *maximal* connected subgraph of $G$ that is *uniformly* labeled. Note that with this definition we can have up to $n$ clusters even when $k = 2$. Given any cluster $C$, we denote by $\partial C$ the subset its vertices adjacent to nodes belonging to other clusters – called *inner border* of $C$. We also denote by $\overline{\partial C}$ the set of all nodes that do not belong to $C$ and are adjacent to at least one node in $\partial C$ – called *outer border* of $C$. Finally, the merging degree $\delta(C)$ of $C$ is then defined as $\delta(C) = \min\{\partial C, \overline{\partial C}\}$. The merging degree of the whole graph $G$, is defined as $\delta(G) = \sum_{C \in \mathcal{P}_y} \delta(C)$, where $\mathcal{P}_y$ is the partition into the above defined clusters induced by $y$.

**Proposition 17** *Given any graph $G$ and any integer $c < n$, there exists a labeling $y$ satisfying $|\delta(G)| \leq 2c$ such that any algorithm makes at least $c$ mistakes.*

**Proof** The proof is a straightforward adaptation of the one of Theorem 2 in Cesa-Bianchi et al. (2009b, 2011). The learning setting used in these two papers is not transductive, i.e., the input graph not known beforehand, in that it is revealed in an incremental fashion. More precisely, let $V_t$ be the subset of all nodes of $V$ observed by the learner until time $t$. During the very first trial $t = 1$, the learner is required to predict the label of an arbitrarily chosen node $v_1 \in V$, and we have $V_1 := \{v_1\}$. Then, at each trial $t = 2, 3, \ldots, n$, it selects a node $q_t \in V_{t-1}$ belonging to the node set of the connected subgraph $G_{t-1} = (V_{t-1}, E_{t-1})$ of $G(V, E)$ *induced* by $V_{t-1}$, where $E_{t-1}$ is therefore the subset of all edges in $E$ connecting any two nodes in $V_{t-1}$ for $t > 2$, while $E_1 := \emptyset$. At any time $t \geq 2$, the learner selects $q_t \in V_{t-1}$ such that there exists at least one node adjacent to it in $V \setminus V_{t-1}$, receives a new vertex $v_t$ adjacent to $q_t$ with all edges connecting it with the nodes in $V_{t-1}$, and is required to output a prediction $\hat{y}(v_t)$ for label $y(v_t) \in \mathbb{R}$, while $V_t := V_{t-1} \cup \{v_t\}$. Then, $y(v_t)$ is revealed and the learner incurs a (real) loss measuring the discrepancy between prediction and true label, which is defined to be equal to $|y(v_t) - \hat{y}(v_t)|$.

The proof of Theorem 2 in Cesa-Bianchi et al. (2009b, 2011) can be easily adapted to our context because it does not exploit the non-transductive nature of that learning setting, and we can view each label as the integer in $[k] = \{1, 2\}$ of the corresponding class,[2] thereby using a $0/1$ loss.

More precisely, following the original proof, let $G_0 = (V_0, E_0)$ be an arbitrarily selected connected subgraph of $G = (V, E)$ such that $|V_0| = n - c$. Let $V'$ be the set of $c$ nodes $V \setminus V_0$. We can choose one arbitrary label in $[k]$ for all the nodes of $V_0$, and force one mistake for the prediction of the label of each node in $V'$. We now need to show that each $\delta(G) \leq 2c$. Note that $G_0$ must be a subgraph of a cluster $C_0 \in \mathcal{P}_y$ determined by the algorithm's predictions. Furthermore, the number of nodes of $C_0$ cannot be smaller than $|V_0| = |V| - c$, which implies $\delta(C_0) \leq \overline{\partial C_0} \leq |V'| = c$. For what concerns the other clusters in $\mathcal{P}_y \setminus C_0$, we have $\sum_{C \in \mathcal{P}_y \setminus C_0} \delta(C) \leq \sum_{C \in \mathcal{P}_y \setminus C_0} \partial C \leq |V'| = c$. Hence, we have $\delta(G) = \delta(C_0) + \sum_{C \in \mathcal{P}_y \setminus C_0} \delta(C) \leq 2c$, thereby concluding the proof. ∎

---

[2]Note that this ensures that the partition induced by $y$ is *regular* according to Section 3 of Cesa-Bianchi et al. (2011), because the difference between any two labels within the same cluster is always smaller (equal to 0) than the one between two labels belonging to two different clusters (equal to 1).

### E.3. Graph families

**Proposition 14** *Let $G$ be a bipartite graph and $\mathcal{H}$ its convex bipartitions. We can learn any labeling in $\mathcal{H}$ with at most 2 mistakes in linear time.*

**Proof** We can find any cut-edge $\{u, v\}$ with at most two mistakes. Using the cut-edge, we infer labels all other nodes $x$ by comparing distances $d(u, x)$ and $d(v, x)$, where $d(\cdot, \cdot)$ is the length of shortest path between two given nodes. ∎

### E.4. Graph families

**Proposition 12** *Let $G$ be chordal. Then, $\omega(G) = h(G)$.*

**Proof** It is well known that $\omega(G) = \text{tw}(G) + 1$ for chordal graphs, in fact, it is one of the ways to define treewidth. Moreover, $h(G) \leq \text{tw}(G) + 1$ as $\text{tw}(H) \leq \text{tw}(G)$ holds for all minors $H$ of $G$ and $\text{tw}(K_c) = c - 1$ for the complete graph on $c$ nodes, see, e.g., Diestel (2017). Combining this with $\omega(G) \leq h(G)$ we get $\omega(G) = h(G)$. ∎

### E.5. Grid graphs

Here we prove Theorem 15 and discuss the GRIDWALKER algorithm in two separate theorems.

Without loss of generality, we assume that the input grid graph $G$ is represented using an $n \times 4$ matrix $M$, where, for all $i \in [n]$, the elements of the $i$-th row are the indices of the nodes adjacent to $i$-th node arranged in a clockwise order, viz., starting from the node positioned at the top in the conventional representation of a grid graph on a plane. In the special case of the nodes with degree smaller than 4, the entries corresponding to nodes that are missing in this representation, are conventially set to be equal to 0.

**Theorem 20** *There exists an algorithm that, operating within the self-directed learning setting, makes not more than $3k$ mistakes for any input grid graph $G(V, E)$ and any convex labeling $y : V \to [k]$, while its time complexity is linear in $n$.*

**Proof** For any subgraph $G'$ of $G$, we denote by $V(G')$ its node set. Given any vertex subset $V' \subseteq V$, we denote by $G(V')$ the subgraph of $G$ induced by all the nodes in $V'$. We call any two clusters $C$ and $C'$ *adjacent* iff there exists an edge $\{u, v\} \in E$ such that $u \in V(C)$ and $v \in V(C')$. We say that a cluster is *discovered* when one of its node labels is revealed for the first time. We prove the theorem by showing a linear time procedure that ensures to discover *all* the clusters of $G$ and, whenever a cluster is discovered, we can infer all its labels by making not more than 3 mistakes.

Given any cluster $C$ of the input grid graph $G$, we say that a node $v$ is at the *corner* of $C$ (corner node) if its degree *in* $C$ is at most 2 if $C$ is not a path graph[3], and is equal to 1 otherwise. Analogously, we say that a node $v$ is at the *border* of $C$ (border node) if its degree *in* $C$ is at most 3 if $C$ is not a path graph, and is equal to 2 otherwise. Hence, each corner node for a cluster $C$ is also one of its border nodes. Finally, we call the *border of $G$*, denoted by $\partial G$, the subgraph of $G$ formed by all nodes with degree at most 3. For the sake of the simplicity of this explanation, we say that

---

[3]Note that if $V(C)$ consists of one node, $C$ is still a path graph.

a node is *visited* if we selected it, predicted its label and received its true label. Finally, we assume that the number of clusters $k$ is larger than 1.

We leverage the following simple property: once we have a border node of a cluster $C$, we can find two opposite corner nodes of $C$, i.e., a pair of corner nodes of $C$ such that the geodesic distance between them is maximal, by making at most 3 mistakes. Finding two opposite corner nodes of $C$ clearly implies that we can infer all the labels assigned to its nodes. We now describe the method to find two opposite corner nodes of any given cluster $C$ making at most 3 mistakes, by starting visiting one of its border node $v$. For now, we assume that $C$ is not a path graph (which also includes the case where $V(C)$ consists of only one node), and we treat the path clusters later separately in the last part of the proof.

If $v$ is a corner node, we choose a node $v'$ adjacent to $v$ in $C$, and we visit one after the other the nodes on the path starting from $v$ and including $v'$, until we visit the first node $u$ that does not belong to $V(C)$, i.e., having a different label[4], or we visit one of the nodes of $\partial G$. Let $u'$ be the node of $V(C)$ adjacent to $u$ in the former case, and the last node visited so far of $C$ in the latter case. $u'$ must be in both cases a corner node of $C$.

Then we proceed by visiting one by one all nodes in the path all contained in $C$ that starts from $u'$ and do not include any other node of the path connecting $v$ with $u'$, until, again, we visit the first node $w$ that does not belong to $C$, or we visit one of the nodes of $\partial G$. Let now $w'$ be the node of $V(C)$ adjacent to $u'$ in the former case, and the last node visited so far of $C$ in the latter case. $w'$ must be in both cases a corner node of $C$. We finally conclude by visiting the path all contained in $C$ starting from $w'$ until we visit the first node $z$ that does not belong to $C$, or we find one of the nodes of $\partial G$. It is immediate to verify that, by the construction of this procedure, the last node visited belonging to $C$ is the corner node opposite to $u'$, so that we can infer all labels of $C$. Furthermore, since the number of cut-edge traversed is at most 3, one for each of the paths described above, the maximum number of mistakes made by predicting each label as equal to the one of the last node visited is 3.

We consider now the case where $C$ is a path graph. If $v \in V(C)$ is the first node visited of $V$, using the above described method we can always predict all labels of $C$ making at most 3 mistakes as follows. We can infer the labels of all nodes of a sub-path of $C$, find one of the two terminal nodes of $C$, make at most two mistakes, and return back to $v$ to visit the remaining part of $C$ until we either visit a node that does not belong to $C$ making one additional mistake, or we visit one of the nodes of $\partial G$. Note that this holds even in the degenerate case where $C$ consists of only one node. If, instead, $v$ is not the first node visited of $V$, we always know that there is a cut-edge between $v$ and some node $u \in V$ which does not belong to $V(C)$ and we can apply the above strategy by visiting a node $v'$ adjacent to $v$ in $C$, i.e, such that the geodesic distance between $z$ and $v$ is exactly equal to 2, and proceeding as in the the case where $C$ is path graph and $v$ is the first node visited of $V$.

It is essential to note that each of the node visited of any cluster $C' \not\equiv C$ must be, by definition, a border node of an adjacent cluster, which allows us to use this procedure to infer the labels of other clusters. Finally, it is also immediate to verify that if we start visiting a node that must be at the border of some clusters, i.e., a node with degree 3 in $G$, at any trial there must be a cluster adjacent to the ones visited so far such that we already visited one of its border nodes, which ensures that keeping applying this procedure we visit all the clusters of $G$, proving the claimed mistake bound.

---

[4]We can also have that $v'$ and $u$ are the same node.

For what concerns the time complexity, if we use (1) a *queue* data structure to enqueue each (border) node visited whenever we discover its cluster, (2) we mark all visited nodes as processed, and (3) we dequeue the new (border) node and start find the labeling of its cluster as described above iff it is not marked as processed, then the worst-case running time of this strategy must be clearly linear in $n$, thereby concluding the proof. ∎

**Theorem 21**  *For any input grid graph $G = (V, E)$, there exists a random convex node labeling $y : V \to [k]$ such that any algorithm $A$ (randomized and deterministic) operating within the self-directed learning setting is forced to make in expectation over the randomization of $y$ at least $k - H_k \geq \frac{k}{4}$ mistakes, where $H_k$ is the $k$-th harmonic number $\sum_{j=1}^{k} \frac{1}{j}$.*

**Proof** Without loss of generality, we assume that $A$ knows the whole partitioning of $G$ into clusters. For any partition of $G$ into $k$ clusters, there are $k!$ possible label assignments. If $y$ is selected uniformly at random from such set of all possible labelings, then $A$ makes in expectation $1 - \frac{i}{k}$ mistakes on the $i$-th cluster discovered for all $i \in [k]$. Summing this quantity over all clusters we obtain $\sum_{i=1}^{k} \left(1 - \frac{i}{k}\right) = k - H_k$, as claimed. ∎

