# OpenReview forum: "Self-Directed Node Classification on Graphs"
_algorithmiclearningtheory.org/ALT/2025/Conference — ALT 2025_

### Official Review · Reviewer_jviV · 2024-11-07
**Solid analysis of new setting for self-directed learning.**

**Rating:** 7
**Confidence:** 3

**Review:**

**Summary**
Self-directed learning is an online learning variant where the learner can adaptively choose the sequence of examples it is presented. The authors study node-classification with convex clusters, where if two nodes have the same label, all nodes on the shortest path between them share the label. They later relax this assumption to approximately convex labelings, and also discuss a simple algorithm for homophilic labelings.

**Pros:**
 - The application of self-directed learning to node classification is well motivated.
 - The authors take the time to carefully introduce all the definitions and notations used.
 - The Good4 algorithm is interesting, and the technical ideas introduced here could be of independent interest. It is also useful that the algorithm works for near-convex labelings with no modifications.

**Cons:**
 - The assumption on convexity is somewhat restrictive, and even with the near-convex case the mistake bound can be bad.

**Questions:**
 - In Proposition 2, it states that $M(Halving, \mathcal{H}) = O(h(G) \ln n)$. Is it the case that $vc(\mathcal{H})=h(G)$? (so it directly follows from the general mistake bound of the halving algorithm?)
 - If I am understanding the definitions correctly, for any set of vertices U, $q(U) = 3{|U| \choose 4}$? So q_good(U) is bounded from below by a function of the size of U, with no dependence on the graph structure?

**Typos:**
 - Algorithms GridWalker and Traverse are mentioned but it does not say where they are defined.
 - Typo in Observation 6: “Then, in any subset U ⊆ V of at least max(w, 4) nodes, there exists an ε-good node, and [the ?] node participating in the biggest number of good quadruples is [an?]ε-good node.”
 - Observation 7: Should this be {(a,b), (c’, d’)} forms a good quadruple?

**Paper Award:**

No

---

> ### Author Response · Authors · 2024-11-21
>
> Thank you for the time and effort you invested in reviewing our paper!
>
> > The assumption on convexity is somewhat restrictive, and even with the near-convex case the mistake bound can be bad.
>
> While we agree that the convexity assumption can be sometimes a bit restrictive, convex communities/clusters are quite common in large real-world networks (see, e.g., Thiessen&Gärtner (2021) and Marc&Šubelj (2018) and the other references mentioned in the paper). In situations where the convexity assumption is not appropriate at all we can instead use our results for the more common homophilic setting (Prop 17, 18).
>
> > In Proposition 2, it states that $M(Halving,\mathcal{H})=O(h(G)\ln n)$. Is it the case that $vc(\mathcal{H})=h(G)$
>
> We will add the fact that $vc=O(h(G))$ to the main body of the paper (it is stated in the proof of Proposition 2 in the appendix). This follows from a result by Duchet & Meyniel (1983) (bounding the so-called Radon number, which bounds the VC-dimension). Obtaining a tight bound (e.g., in terms of different graph parameters) is still open. A somewhat general lower bound is $vc=\Omega(\omega(G))$ for so-called $S_4$ graphs, where $\omega(G)$ is the clique number. Nevertheless, for many graphs the bound is tight (e.g., bounded treewidth graphs, planar graphs, $S_4$ chordal graphs, etc.). See, e.g., the cited Chepoi (2024) survey for some overview on the progress (again mostly in terms of the Radon number). We will add and clarify these points in the final version.
>
>
> >If I am understanding the definitions correctly, for any set of vertices U, $q(U)=3{|U| \choose 4}$? So q_good(U) is bounded from below by a function of the size of U, with no dependence on the graph structure?
>
> No, arbitrary sets $U$ could have no good quadruples at all. For example, clique graphs have no good quadruples (no shortest paths overlap internally). Hence in general, there is no lower bound on $q_\mathrm{good}(U)$. The dependence on the graph is more explicit in, e.g, Proposition 3. Here, we require $U$ to have size at least $\max(w,4)$ where $h(G)<w$; only then we can guarantee the existence of good quadruples (and a lower bound on $q_\mathrm{good}(U)/q(U)$).
>
> > Typos:
>
> Thank you, we will fix the typos. GridWalker and Traverse are defined in the appendix, we will clarify this in the final version.

---

### Official Review · Reviewer_tTj4 · 2024-11-08
**A decent paper providing meaningful insights into an interesting learning-algorithmic problem**

**Rating:** 7
**Confidence:** 4

**Review:**

This paper is the first to study self-directed learning for the classification of vertices in a given graph. The underlying assumption is that the (connected) graph is fully given, and that each class of vertices forms a convex cluster. Initially, it is assumed that there are only 2 clusters. The authors designed a self-directed learning algorithm that runs in polynomial time in the number n of vertices and whose number of mistakes is upper-bounded by a function logarithmic in n and polynomial in the Hadwiger number of the given graph.

Some variations on the problem are discussed. For example, when convexity of clusters might be violated, the mistake bound of the proposed algorithm increases by at most 4 times the smallest number of vertices to be moved into the opposite cluster so as to fulfil the convexity criterion. In addition, for learning special classes of graphs, a few simple observations can be made on the number of mistakes. The multi-cluster case is also discussed (mainly in the appendix). Finally, it is observed (again with a very simple argument) that efficient self-directed learning is always possible with at most as many mistakes as there are boundary vertices (on the boundary of the clusters).

I consider the observations made from Theorem 9 onward fairly straightforward; the main result is clearly Theorem 8. However, I still think that the paper is worth publishing, and I am not a fan of requiring big muscle-flexing in order to have a paper accepted. The proposed algorithm makes use of structural properties of the underlying graph (sparsity in the sense of lack of K_z minors except for small values of z); these insights might be helpful for others working on learning graphs.

Still, some issues bothered me a bit:
(1) I appreciated the discussion of potential application domains, but was wondering when we can really assume a small Hadwiger number. For example, in social networks, might it not happen that the Hadwiger number becomes large enough to make the mistake bound large?
(2) Secondly, the authors state that their algorithm runs in polynomial time, which is certainly true. However, it needs to calculate shortest paths between lots of pairs of vertices, which, in large graphs like social networks may not really be feasible. I was wondering whether the authors have looked at the problem of trying to bring down the computational cost of their algorithm as far as possible. Perhaps a lot can be shaved off in terms of runtime at only a small cost in terms of mistakes of the self-directed predictor?
(3) For domains in which the Hadwiger number is prohibitively large, can one perhaps still salvage the algorithmic idea if one assumes that all vertices that belong to a large clique are in the same cluster? Such assumption may be fairly realistic in social networks, perhaps. I think this could lead to an interesting extension of the results presented in this paper.

Some minor comments for improvement of the writing:
- The term "$\varepsilon$-good" suggests that $\varepsilon$ is an external parameter that can be set independently of everything else. But, if I understood correctly, $\varepsilon$ is determined by the given subset $U$ of the vertex set. Would it be more appropriate to call a vertex $U$-good instead of $\varepsilon$-good, and then simply call it good if $U$ is clear from the context? I also noticed that the authors go back and forth between the terms \emph{good} and \emph{$\varepsilon$-good}.
- In line 6 of page 8, it says "indicating that no mistakes have been made yet." technically, shouldn't it say "indicating that no mistakes have been made yet (except possibly when predicting the label of $a$)."?
- line 5 of Section 1: total number *of* classes
- overfull hbox on p 4
- p 5, last line: delete "the" before $U$
- in Def 5, $\varepsilon(U)$ is introduced and never used; elsewhere it just says $\varepsilon$.
- Observation 6 is hard to parse, probably due to syntax mistakes. I think the authors wanted to write "and *any* node participating in the biggest number of good quadruples is *an* $\varepsilon$-good node."
- 4 lines above Thm 8: when *the* induced
- 1 line below Thm 8: to *an* algorithm
- 5 lines below Thm 8: that *an* $\varepsilon$-good node *exists*
- 6 lines below Thm 8: "such as" should be "such that"
- p 8 line -3: among *the* first; also delete "an" after "at least"
- p 8, line -2: delete "among first" or fix grammar
- p 9, line 1: will *make* no mistake
- line after Prop 12: circular-arc *graphs*
- references often have capitalization missing in journal names and conference names.
- p 21, last line: at least *an*
- p 27, line -5: We *predict*

**Paper Award:**

No

---

> ### Author Response · Authors · 2024-11-21
>
> Thank you for your rigorous assessment and helpful suggestions!
>
> > (1) I appreciated the discussion of potential application domains, but was wondering when we can really assume a small Hadwiger number. For example, in social networks, might it not happen that the Hadwiger number becomes large enough to make the mistake bound large?
>
> We agree that it is not absolutely certain that social networks have a small Hadwiger number. Nevertheless, we chose the Hadwiger number as it is a common parameter for graph sparseness (Nešetřil&De Mendez, 2012; Demaine et al., 2019) and there are solid empirical reasons justifying it. Social networks have dense communities, but they are globally rather sparse. They are often modeled as scale-free graphs following the power-law distribution (see, e.g., [1,2]), supporting a small Hadwiger number. Also such networks often arise from hyperbolic metric spaces (see, e.g., [3,4]). As mentioned in our paper some of these hyperbolic random graph models tend to produce graphs with small Hadwiger number (Bläsius et al., 2016). We will add this discussion to the next version.
>
> [1] Barabási, Albert-László, and Réka Albert. "Emergence of scaling in random networks." Science 286.5439 (1999): 509-512.
> [2] Girvan, Michelle, and Mark EJ Newman. "Community structure in social and biological networks." Proceedings of the national academy of sciences 99.12 (2002): 7821-7826.
> [3] Papadopoulos, Fragkiskos, et al. "Greedy forwarding in dynamic scale-free networks embedded in hyperbolic metric spaces." Proceedings IEEE Infocom. IEEE, 2010.
> [4] Chen, Yankai, et al. "Modeling scale-free graphs with hyperbolic geometry for knowledge-aware recommendation." Proceedings of the fifteenth ACM international conference on web search and data mining. 2022.
>
> > (2) Secondly, the authors state that their algorithm runs in polynomial time, which is certainly true. However, it needs to calculate shortest paths between lots of pairs of vertices, which, in large graphs like social networks may not really be feasible. I was wondering whether the authors have looked at the problem of trying to bring down the computational cost of their algorithm as far as possible. Perhaps a lot can be shaved off in terms of runtime at only a small cost in terms of mistakes of the self-directed predictor.
>
> This is an excellent point. A major bottleneck is, as you say, the all-pairs-shortest-path problem, i.e., computing the full distance matrix (typically with runtime $\Theta(|V|^3)$ in the worst-case). One possibility could be to estimate the distance matrix (or just the distances currently required) by sampling a small set of landmarks $L$: $d(a,b) \le \min_{\ell\in L} d(a,\ell) + d(\ell,b)$. The impact of such sampling approaches on our mistake bound is unclear, making it an excellent direction for future research.
>
> > (3) For domains in which the Hadwiger number is prohibitively large, can one perhaps still salvage the algorithmic idea if one assumes that all vertices that belong to a large clique are in the same cluster? Such an assumption may be fairly realistic in social networks, perhaps. I think this could lead to an interesting extension of the results presented in this paper.
>
> Thanks for this suggestion; we agree. Indeed various related convexity-based assumptions exist (e.g., taking arbitrary paths or induced paths instead of shortest paths). We started with the geodesic assumption as it is the most natural and the most studied once. Also, more common homophilic assumptions typically lead to monochromatic cliques (or at least not large cliques split evenly, as these would induce a very large cutsize). Maybe a combination of homophilic and geodesic assumptions is reasonable.
>
> > $\varepsilon$-good
>
> Good point, we will change this in the final version if our manuscript is accepted.
>
> > further minor comments
>
> Thanks for all these points, we will fix them!

---

### Official Review · Reviewer_DbUU · 2024-11-10
**This paper provides a poly-time algorithm for self-directed node classification with at most O(h(G)^4 * ln n) mistakes. Given the innovation, clarity, and potential for impactful applications, I recommend this paper for acceptance.**

**Rating:** 7
**Confidence:** 3

**Review:**

Summary:

This paper considers the problem of self-directed node classification in graphs, where the learner can select a node to classify in each round. This paper mainly considers the unknown labeling of nodes is a convex bipartition (or halfspace) and also generalizes to the nearly convex and convex k labeling. Here, the labeling is called convex iff for any two nodes $u,v$ have the same label, the nodes on the shortest path between u and v have the same label. Ben-David et al. (1997) showed that the halving algorithm makes $O(h(G) \ln n)$ mistakes for convex bipartition. However, this algorithm is not computationally efficient. Here, $h(G)$ the Hadwiger number of $G$, which is the size of the largest clique minor in $G$.

The paper introduces a polynomial time algorithm, GOOD4, which is shown to achieve a mistake bound of $3(h(G)+1)^4 \ln n$ for graphs with two convex clusters. The algorithm is also robust in cases where clusters are nearly convex. The labeling is $M^*$ nearly convex if by flipping at most $M^*$ labels of nodes, the labeling can be converted to convex. They show that this algorithm achieves a $3(h(G)+1)^4 \ln n + 4M^*$ mistake bound. They also provide an algorithm for the multi-class classification.  Additionally, the paper presents lower bounds on the number of mistakes for certain graph families and explores the problem in the context of homophilic labeling.

Pros:

(1) The proposed algorithm, GOOD4, achieves polynomial runtime with mistake bounds that scale well, particularly for graphs with bounded Hadwiger numbers. The algorithm is surprisingly easy to implement and achieves a good mistake bound (close to the previous bound by the halving algorithm, which is not efficient. ).

(2) The algorithm’s robustness to nearly convex labelings, with mistake bounds that account for minor deviations from convexity, enhances its applicability. The paper also covers extensions to more complex 𝑘 k-labeling scenarios, expanding the relevance of the algorithm to a wider range of labeling structures.

(3) The analysis of lower bounds on mistakes for various graph families provides the theoretical understanding of self-directed node classification.

Cons:

(1) The mistake bound achieved by GOOD4, $3(h(G)+1)^4 \ln n$, is significantly higher than the $O(h(G) \ln n)$ bound of the halving algorithm, particularly for graphs where the Hadwiger number $h(G)$ is large. This difference stems from the $h(G)^4$ factor in GOOD4’s bound, which can lead to considerably more mistakes in graphs with high Hadwiger number, $h(G)$.

Minors:

(1) Algorithm Line 18: It should be if $\hat{y}_c \neq y(c)$.

(2) Proposition 11: two learning typo

(3) Appendix A, Observation 7 proof, use definition 5 instead of observation 7.

**Paper Award:**

No

---

> ### Author Response · Authors · 2024-11-21
>
> Thank you for your detailed and constructive feedback!
>
> > (1) The mistake bound achieved by GOOD4, $3(h(G)+1)^4\ln n$, is significantly higher than the $O(h(G)\ln n)$ bound of the halving algorithm, particularly for graphs where the Hadwiger number is large $h(G)$. This difference stems from the factor in GOOD4’s bound, which can lead to considerably more mistakes in graphs with high Hadwiger number, $h(G)$.
>
> The mistake bound of Good4 is indeed asymptotically larger than Halving's. However, we believe that the $O(h(G)\ln n)$ mistake bound is not achievable in polynomial time. For example, it is unlikely that Halving can be implemented in polynomial time for our hypothesis space of graph halfspaces (even just checking if a halfspace consistent with a given sample exists is NP-hard). Closing this gap and finding the exact rate achievable within polynomial time constraints remains an area for future research.
>
>
> >Minors: ...
>
> Thanks for spotting, we will fix them!

---

### Author Rebuttal · Authors · 2024-11-21

We thank all reviewers for their time and the encouraging and detailed reviews. We will comment individually below.

---

### Meta-Review · Area_Chair_zpP5 · 2024-12-13

**Recommendation:** Accept
**Confidence:** 4

**Metareview:**

The main result is an algorithm for self-directed classification of convex sets of vertices in a graph with bounded hadwiger number h (convex set = for u, v in set all vertices on a shortest u-v path are also in the set; hadwiger number = size of the largest complete minor contained in the graph; self-directed classification = algorithm chooses the vertices in the graph to label/get feedback). The algorithm runs in polynomial time and achieves poly(h)log(n) mistakes. The reviewers all agree that this is a solid paper; I read the paper and agree with the assessment.

Strengths: natural algorithm, clean analysis, the algorithm is robust (allowing deviation from the convexity assumption)

Weaknesses: convexity is a bit restrictive, running time (while polynomial) is rather large, the assumption of having bounded hadwiger number needs more justification (the authors provided more justification in the response to reviewers).

**Paper Award:**

No